# IMPACTS OF CLIMATE CHANGE ON GROUNDWATER FLOODING AND ECOHYDROLOGY IN LOWLAND KARST

*(Re-submitted 15/02/2021)*

**Patrick Morrissey[1]**, Paul Nolan[3], Ted McCormack[2], Paul Johnston[1], Owen Naughton[2], and Laurence Gill[1]

[1] Department of Civil, Structural & Environmental Engineering, Trinity College Dublin, University of Dublin, Museum Building, College Green, Dublin 2, Ireland
2 Geological Survey of Ireland, Beggars Bush, Haddington Road, Dublin 4, Ireland
3 Irish Centre for High-End Computing (ICHEC), 2, 7/F, The Tower, Trinity Technology & Enterprise Campus, Grand Canal Dock, Dublin 2, Ireland

*Corresponding Author: Patrick Jerome Morrissey, Email Address: morrispj@tcd.ie*

*Keywords:* Climate change, groundwater flooding, karst hydrology, karst flooding, eco-hydrology

## Abstract

Lowland karst aquifers can generate unique wetland ecosystems which are caused by groundwater fluctuations that result in extensive groundwater-surface water interactions (i.e. flooding). However, the complex hydrogeological attributes of these systems linked to extremely fast aquifer recharge processes and flow through well-connected conduit networks often present difficulty in predicting how they will respond to changing climatological conditions. This study investigates the predicted impacts of climate change on a lowland karst catchment by using a semi-distributed pipe-network model of the karst aquifer populated with output from the high spatial resolution (4 km) COSMO-CLM regional climate model simulations for Ireland. An ensemble of projections for the future Irish climate were generated by downscaling from five different global climate models (GCMs), each based on four Representative Concentration Pathways (RCP2.6, RCP4.5, RCP6.0 and RCP8.5) to account for the uncertainty in the estimation of future global emissions of greenhouse gases. The one dimensional hydraulic / hydrologic karst model incorporates urban drainage software to simulate open channel and pressurised flow within the conduits with flooding on the land surface represented by storage nodes with the same stage-volume properties of the physical turlough basins. The lowland karst limestone catchment is located on the west coast of Ireland and is characterised by a well-developed conduit dominated karst aquifer which discharges to the sea via intertidal and submarine springs. Annual above ground flooding associated with this complex karst system has led to the development of unique wetland ecosystems in the form of ephemeral lakes known as turloughs, however extreme flooding of these features causes widespread damage and disruption in the catchment. This analysis has shown that mean, 95th and 99th percentile flood levels are expected to increase by significant proportions for all future emission scenarios. The frequency of events currently considered to be extreme is predicted to increase, indicating that more significant groundwater flooding events seem likely to become far more common. The depth and duration of flooding is of extreme importance, both from an ecological perspective in terms of wetland species distribution and for extreme flooding in terms of the disruption to homes, transport links and agricultural land inundated by flood waters. The seasonality of annual flooding is also predicted to shift later in the flooding season which could have consequences in terms of ecology and land use in the catchment. The investigation of increasing mean sea levels, however showed that anticipated rises would have very little impact on groundwater flooding due to the marginal impact on ebb tide outflow volumes. Overall, this study highlights the relative vulnerability of lowland karst systems to future changing climate conditions mainly due to the extremely fast recharge which can occur in such systems. The study presents a novel and highly effective methodology for studying the impact of climate change in lowland karst systems by coupling karst hydrogeological models with the output from high resolution climate simulations.
.

# Introduction

Climate projections indicate that a shift in the magnitude and pattern of precipitation is likely to alter catchment runoff regimes in Ireland (Nolan et al., 2017, Blöschl et al., 2019, Murphy et al., 2019). As a consequence, extreme events, such as floods and droughts, are expected to increase in frequency and intensity (Noone et al., 2017, Blöschl et al., 2019). These predicted changes in precipitation will undoubtedly impact groundwater resources and groundwater-related phenomena such as groundwater flooding and groundwater-dependent wetland habitats. Many studies have previously attempted to postulate the likely impacts of climate change on groundwater resources without using a combination of numerical models driven by climate data derived from Global Climate Models (GCM) (Dragoni and Sukhija, 2008, Howard and Griffith, 2009, Taylor et al., 2013, Meixner et al., 2016). These studies also tend to focus on groundwater resources in terms of the provision of a potable water supply or irrigation and so have not been considered groundwater flooding or eco-hydrology in detail. They have also not been focused on groundwater systems dominated by karst flow Studies into the impacts of climate change have been carried out for the chalk aquifers of south-western England which have high porosity and are prone to karstification. Jackson et al. (2015) utilised a distributed ZOOMQ3D groundwater model of the Chalk aquifer with various emission scenario input data to investigate the predicted changes in groundwater levels. Brenner et al. (2018) conducted a further study of this chalk catchment and showed that projected climate changes may lead to generally lower groundwater levels and a reduction of exceedances of high groundwater level percentiles in the future. Chen et al. (2018) conducted a study into the effects of climate change on alpine karst using GCM data. However, the results of these studies are not directly relevant to lowland karst with significant groundwater-surface water interactions and associated eco-hydrological habitats (groundwater fed wetlands). In order to assess the future risks relating to groundwater flooding and eco-hydrology in lowland karst, it is imperative to understand the complex hydrological processes governing groundwater flow in karst bedrock and how it will likely be altered in the future (Morrissey et al., 2020). In this context, various forms of numerical models are usually applied to describe the hydrological processes in karst catchments (Fleury et al., 2009, Gill et al., 2013a, Hartmann et al., 2013, Hartmann, 2017, Mayaud et al., 2019), which can accurately simulate the groundwater flow and flooding processes which typically occur. Global and distributed models have been successfully applied to simulate lowland karst with lumped models typically favoured due to their ease of use in gauged catchments. When considering eco-hydrology (specifically Groundwater Dependent Terrestrial Ecosystems – GWDTE), droughts and extreme floods present the greatest climatological threat and therefore the impacts of predicted climate change are of immediate concern. Whilst fluvial models (models which simulate flow with rivers) are relatively straightforward to calibrate and couple with the output from Global or Regional Climate Models, groundwater (and specifically karst) models can be more difficult to employ in such a manner, particularly in terms of assessing the resultant output (Hartmann, 2017). Predicting extreme values with limited gauging data follows established well validated methodologies (Griffis and Stedinger, 2007, Shaw et al., 2011, Ahilan et al., 2012) and; however no such established methods appear to be available currently for groundwater flooding in karst systems.

The phenomenon of groundwater flooding in general has become more reported as a natural hazard in recent decades following extensive damage to property and infrastructure across Europe in the winter of 2000-2001 (Finch et al., 2004, Pinault et al., 2005, Hughes et al., 2011). Significant groundwater flooding also occurred in the UK at Oxford (2007) and at Berkshire Downs and Chilterns (2014) and in Galway, Ireland in 2009 & 2015/2016 (Naughton et al., 2017). Groundwater flooding occurs when the water table rises above the land surface flooding areas often for prolonged periods (often many weeks or months). This compares to fluvial flooding which occurs when river (or lake) systems overflow their banks and flow into the surrounding lands. Fluvial flooding typically occurs in a sudden (or dramatic) and sometimes dangerous manner following intense rainfall and dissipates relatively quickly (days). Whilst it has been reported that groundwater flooding rarely poses a risk to human life, this form of flooding is known to cause damage and disruption over a long duration, particularly when compared to fluvial flooding (Morris et al., 2008, Cobby et al., 2009). Climate change is also likely to further exacerbate extreme droughts (Murphy et al., 2019) and their frequency and persistence must be quantified if resource planning and protection are to be implemented. Equally, as discussed, the effects of changes in hydrological regimes to wetland ecosystems can be significant; for example, recent studies (Spraggs et al., 2015, Noone et al., 2017) have attempted to quantify the frequency and extent of historic droughts

to better understand their recurrence interval and thus assess the resilience of different impacted wetland
ecosystems. Hence, this study aims to assess the predicted impacts of climate change, particularly
during these extreme events, using an ensemble of Regional Climate Models to provide input data into
a semi-distributed model of a lowland karst catchment in the West of Ireland.
The impact of increasing greenhouse gases and changing land use on climate change can be simulated
using Global Climate Models (GCMs). However, long climate simulations using GCMs are currently
feasible only with horizontal resolutions of ~50 km or coarser. Since climate fields such as precipitation,
wind speed and temperature are closely correlated to the local topography, this is inadequate to simulate
the detail and pattern of climate change and its effects on the future climate of Ireland. Hence, Regional
Climate Models (RCMs) have been developed by dynamically downscaling the coarse information
provided by the global models to provide high-resolution information on a subdomain covering Ireland.
The computational cost of running the RCM, for a given resolution, is considerably less than that of a
global model. The approach has its flaws; all models have errors, which are cascaded in this technique,
and new errors are introduced via the flow of data through the boundaries of the regional model.
Nevertheless, numerous studies have demonstrated that high-resolution RCMs improve the simulation
of fields such as precipitation (Kendon et al., 2012, Lucas-Picher et al., 2012, Kendon et al., 2014,
Bieniek et al., 2016) and topography-influenced phenomena and extremes with relatively small spatial
or short temporal character (Feser et al., 2011, Feser and Barcikowska, 2012, Shkol'nik et al., 2012,
IPCC, 2013). The physically based RCMs explicitly resolve more small-scale atmospheric features and
provide a better representation of convective precipitation (Rauscher et al., 2010) and extreme
precipitation (Kanada et al., 2008). Other examples of the added value of RCMs include improved
simulation of near-surface temperature (Feser, 2006, Di Luca et al., 2016), European storm damage
(Donat et al., 2010), strong mesoscale cyclones (Cavicchia and von Storch, 2012), North Atlantic tropical
cyclone tracks (Daloz et al., 2015) and near-surface wind speeds (Kanamaru and Kanamitsu, 2007),
particularly in coastal areas with complex topography (Feser et al., 2011, Winterfeldt et al., 2011). The
IPCC have concluded that there is "high confidence that downscaling adds value to the simulation of
spatial climate detail in regions with highly variable topography (e.g., distinct orography, coastlines) and
for mesoscale phenomena and extremes" (IPCC, 2013).

## Study Catchment

Groundwater flooding in Ireland predominantly occurs within the lowland limestone areas of the west of
the country (Naughton et al., 2012, Naughton et al., 2018). This flooding is governed by complex
interactions between ground and surface waters, with sinking and rising rivers/streams common and
surface water features absent completely in many areas (Drew, 2008). The flooding is controlled by
complex geology whereby the dominant drainage path for many catchments is through the karstified
limestone bedrock. During intense or prolonged rainfall, the limestone bedrock is unable to drain
recharge due to the limited storage available within the bedrock (fractures and conduits). Turloughs occur
in glacially formed depressions in karst, which intermittently flood on an annual cycle via groundwater
sources and have substrate and/or ecological communities characteristic of wetlands.
Geomorphologically they are a variant on a polje which are generally larger and more flat-bottomed
enclosed depressions in karst landscapes (Ford and Williams, 2007). In Ireland, the most susceptible
region to groundwater flooding is the south Galway Lowlands, centred around the town of Gort, which is
a lowland karst catchment covering an area of approximately 500 km$^2$ (Naughton et al., 2018).
The lowland karst catchment is made up of two distinct bedrock geologies with the upland mountainous
areas to the east underlain by Old Red Sandstone and the lowlands in the west underlain by highly
permeable karstified Carboniferous Limestone. The presence of a permeable epikarst with a well-
developed conduit and cave system dispersed throughout the limestone portion of the catchment has
given rise to a very distinct surface hydrology which large volumes of water exchanged between the
surface and subsurface across the lowlands through sinking streams, large springs and estavelles
(Naughton et al., 2018). Three rivers flow off the Slieve Aughty Mountains (much of which are covered
in blanket bog and forestry) providing allogenic recharge into the lowland karst and a fourth flows into
the catchment from the south-west. Once these watercourses contact the limestone they disappear into

the bedrock where flow occurs within caves or conduits – see Figure 1. The rivers reappear for short intervals at a number of locations before discharging to the sea via submarine groundwater discharge (including springs located at the intertidal zone of the bay) at Kinvara Bay (Gill et al., 2013b). The groundwater conduit network surcharges to the ground surface through estavelles and springs following periods of sustained heavy rainfall when sufficient capacity is not available in the bedrock to store and convey water to the sea. The excess surface water floods turloughs and interconnected floodplains across the catchment. Extensive and damaging flooding associated with these turloughs has occurred twice in the last decade leading to considerable cost and disruption. An extreme flood event which occurred in November 2009 was the most severe on record, until it was surpassed in many areas by the events of 2015/2016. These floods led to over 24 km$^2$ of land being inundated for up to 6 months. The apparent increase in frequency with which these hugely damaging extreme flooding events are occurring has made quantifying the likely impact of future climate change a topic of high priority and importance. In addition, given that the entire catchment drains to a series of springs at the coast (some of which are intertidal) the impacts of rising sea level, either in combination or isolation to changing rainfall patterns associated with climate change, are also of concern.

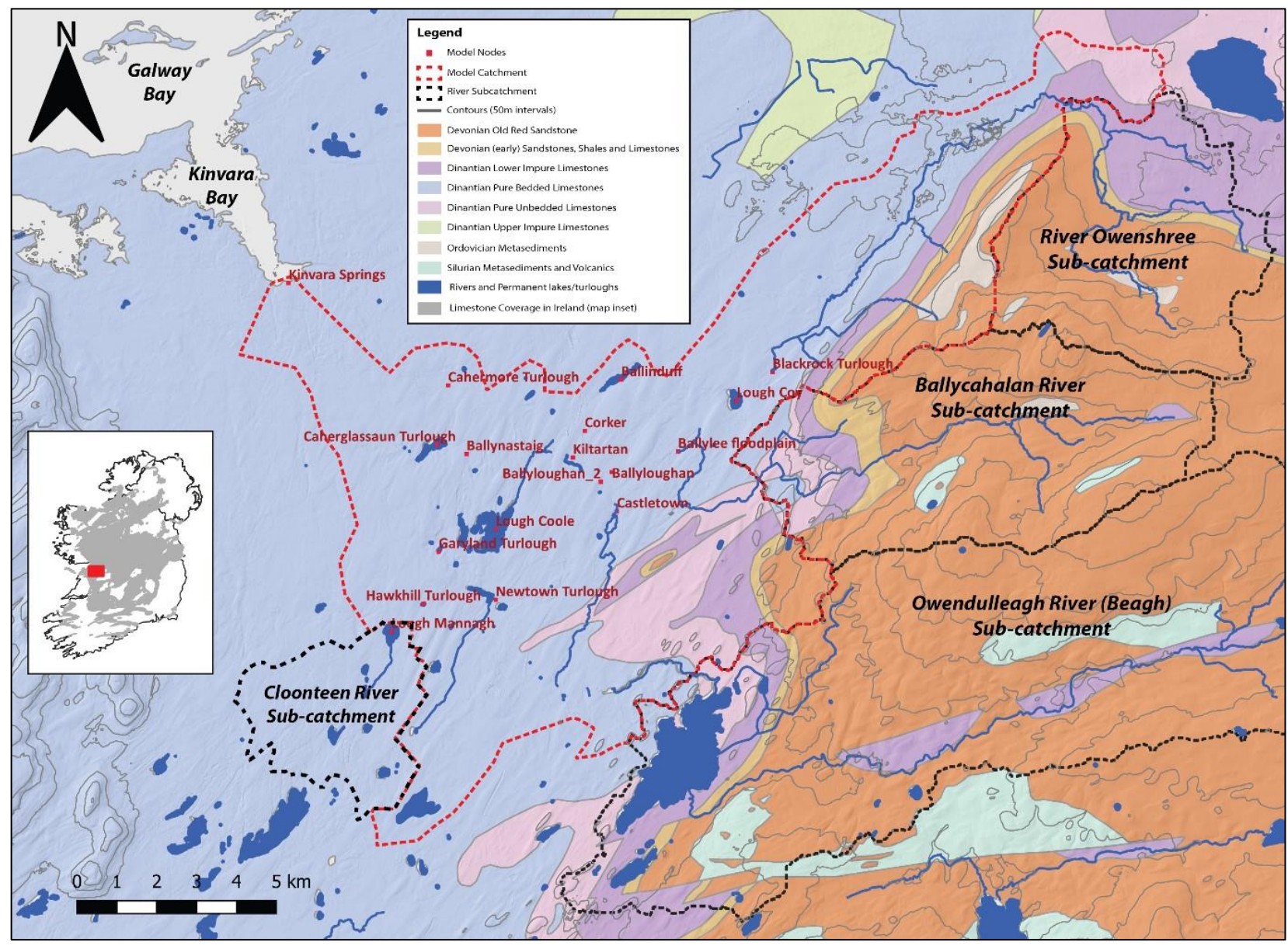

*Figure 1 Map of the catchment showing geology, major rivers/lakes and nodes within the model catchment*

## Methodology

**Climate Models and Methods**

The future climate of Ireland was simulated at high spatial resolution (4 km) using the COSMO-CLM (v5.0) RCM. The COSMO-CLM regional climate model is the COSMO weather forecasting model in climate mode (www.clm-community.eu, Rockel et al., 2008). The COSMO model (www.cosmo-model.org) is the non-hydrostatic operational weather prediction model used by the German Weather Service (DWD). Projections for the future Irish climate were generated by downscaling the following CMIP5 global datasets; the UK Met Office's Hadley Centre Global Environment Model version 2 Earth System configuration (HadGEM2-ES) GCM, the EC-Earth consortium GCM, the CNRM-CM5 GCM developed by CNRM-GAME (Centre National de Recherches Météorologiques—Groupe d'études de l'Atmosphere Météorologique) and Cerfacs (Centre Européen de Recherche et de Formation Avancée), the Model for Interdisciplinary Research on Climate (MIROC5) GCM developed by the MIROC5 Japanese research consortium and the MPI-ESM-LR Earth System Model developed by the Max Planck Institute for Meteorology. The Representative Concentration Pathways (RCPs) are greenhouse gas concentration trajectories adopted by the IPCC. The RCPs are focused on radiative forcing – the change in the balance between incoming and outgoing radiation via the atmosphere caused primarily by changes in atmospheric composition – rather than being linked to any specific combination of socioeconomic and technological development scenarios. There are four such scenarios (RCP2.6, RCP4.5, RCP6.0 and RCP8.5), named with reference to a range of radiative forcing values for the year 2100 or after, i.e. 2.6, 4.5, 6.0 and $8.5W/m^2$, respectively (Moss et al., 2010; van Vuuren et al., 2011).

The RCMs were driven by GCM boundary conditions with the following nesting strategy; GCM to 18 km and GCM to 4 km. For the current study, only 4 km grid spacing RCM data are considered. The higher resolution data allows sharper estimates of the regional variations of climate projections. The climate fields of the RCM simulations were archived at 3-h intervals.

The mid-century precipitation climate of Ireland is expected to become more variable with substantial projected increases in both dry periods and heavy precipitation events (Nolan 2017, 2020). These studies show that substantial decreases in precipitation are projected for the summer months, with reductions ranging from 0% to 11% for the RCP4.5 scenario and from 2% to 17% for the RCP8.5 scenario. Other seasons, and over the full year, show relatively small projected changes in precipitation. The frequencies of heavy precipitation events show notable increases over the year as a whole and in the winter and autumn months, with projected increases of 5–19%. The number of extended dry periods is also projected to increase substantially by the middle of the century over the full year and for all seasons except spring. The projected increases in dry periods are largest for summer, with values of +11% and +48% for the RCP4.5 and RCP8.5 scenarios, respectively. Refer to Figure 2 for further details.

An overview of the simulations is presented in Table 1. Data from two time-slices, 1976–2005 (the control or past) and 2071–2010, were used for analysis of projected changes in the Irish climate by the end of the 21st-century. It must be noted that the full RCM simulations in fact covered the entire period 1976 – 2100 and these time slices were simply used to make a past versus future comparison (Figure 2 shows results from the full simulation and not just the chosen time slices for this current study). The historical period was compared with the corresponding future period for all simulations within the same RCM-GCM group. This results in future anomalies for each model run; that is, the difference between future and past.

*Table 1: Details of the ensemble RCM simulations used in this study; rows present information*
*on the RCM used, the corresponding downscaled GCM, the RCP used for future simulations,*
*the number of ensemble comparisons and the time-slice analysed. In each case, the future 30-*
*year period 2071 - 2100 are compared with the past RCM period 1976-2005. the mean of three*
*RCP2.6, five RCP4.5 and five RCP8.5 RCM projections were calculated. The RCP6.0 simulation*
*comprises just one simulation so was compared directly with the past RCM period.*

| RCM | GCM | Scenarios | No. of ensemble comparisons | Time periods analysed |
|---|---|---|---|---|
| | EC-Earth (r1i1p1) | Historical RCP4.5, RCP8.5 | - 2 | 1976 – 2005 2071 - 2100 |
| | MPI-ESM-LR (r1i1p1) | Historical RCP2.6, RCP4.5, RCP8.5 | - 3 | 1976 – 2005 2071 - 2100 |
| COSMO5 | CNRM-CM5 (r1i1p1) | Historical RCP4.5, RCP8.5 | - 2 | 1976 – 2005 2071 - 2100 |
| | HadGEM2-ES (r1i1p1) | Historical RCP2.6, RCP4.5, RCP8.5 | - 3 | 1976 – 2005 2071 - 2100 |
| | MIROC5 (r1i1p1) | Historical RCP2.6, RCP4.5, RCP6.0, RCP8.5 | - 4 | 1976 – 2005 2071 - 2100 |


The RCM projection results are in line with previous work (McGrath et al., 2005; McGrath and
Lynch, 2008, Gleeson et al., 2013, Nolan et al., 2014, 2017, 2020, Nolan, 2015, O'Sullivan et
al., 2015) with enhanced temperature rises predicted by the end-of-century of between 0.8 to
3ºC for the high emission scenario (RCP8.5).

The method of bilinear interpolation was employed to extract 5 km RCM precipitation and
evapotranspiration data at each of the locations of existing rain gauges in the study catchment.
The Penman-Monteith FAO-56 method (REF) was used to compute daily evapotranspiration
(mm) (see Werner et al. 2018 for a full description of methods and validations).

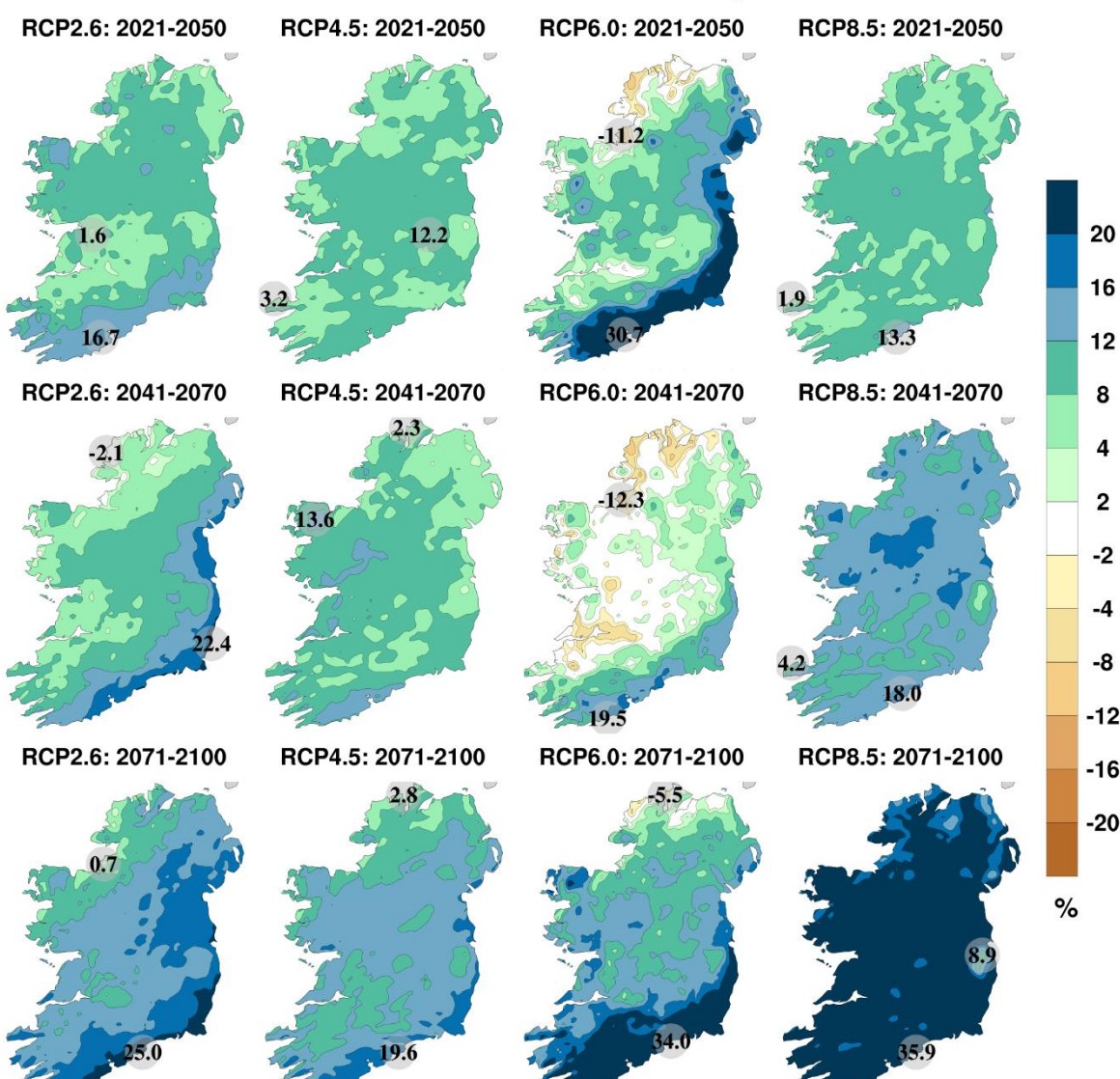

**Figure 2: RCM Ensemble Projections of Mean Winter Rainfall (%). The individual ensemble percentage projections are calculated as 100×(future-past)/past. In each case, the future 30-year periods are compared with the past RCM period 1976-2005. The figure presents the mean of three RCP2.6 (Low), five RCP4.5 (Med), one RCP6.0 (Med/High) and five RCP8.5 RCM (High) projections. The numbers included on each plot are the minimum and maximum projected changes, displayed at their locations. (refer to Figure 1 for location of study catchment)**

The RCMs were validated by downscaling ECMWF ERA-Interim reanalyses and the GCM datasets for multi-decadal time periods and comparing the output with observational data. Extensive validations were carried out to test the ability of the RCMs to accurately model the climate of Ireland.  (a) presents the annual observed precipitation averaged over the period 1981–2000. Figure 3 (b) presents the downscaled ERA-Interim data as simulated by the COSMO5-CLM model with 4-km grid spacings. It is noted that the RCM accurately captures the magnitude and spatial characteristics of the historical precipitation climate, e.g. higher rainfall amounts in the west and over mountains.

Figure 3 (c) shows that the percentage errors range from approximately –30% to
approximately +15% for COSMO5-CLM downscaled ERA-Interim data. The percentage error
at each grid point (i, j) is given by:

$per\_bias_{(i,j)} = 100 \times \left( \dfrac{bias_{(i,j)}}{\overline{OBS}_{(i,j)}} \right)$     *(Eq.1)*
*where*
$$bias_{(i,j)} = \overline{RCM}_{(i,j)} - \overline{OBS}_{(i,j)} \qquad (Eq.2)$$

and the $\overline{RCM}_{(i,j)}$ and $\overline{OBS}_{(i,j)}$ terms represent the RCM and observed values, respectively, at
grid point (i, j), averaged over the period 1981–2000. Figure 3 (c) highlights a clear
underestimation of precipitation over the mountainous regions. This is probably because the
RCMs underestimate heavy precipitation; previous validations studies (e.g. Nolan et al., 2017)
have demonstrated a decrease in RCM skill with increasing magnitude of heavy precipitation
events.
To assess the added value of high-resolution RCM data, and to quantify the improved skill of
RCMs over the GCMs, precipitation data were compared with both RCM and GCM data for
the period 1976–2005. Results, presented in Table 2, demonstrate improved skill of the RCMs
over the GCMs. Moreover, an increase in grid resolution of the RCMs (from 18- to 4-km grid
spacings) results in a general increase in skill.
For an in-depth validation of the RCMs, please refer to Nolan et al. (2015, 2017, 2020),
Flanagan et al. (2019, 2020) and Werner et al. (2019), the results of which confirm that the
output of the RCMs exhibit reasonable and realistic features as documented in the historical
data record and consistently demonstrate improved skill over the GCMs. The results of these
validation analyses confirm that the RCM configurations and domain size of the current study
are capable of accurately simulating the climate of Ireland.

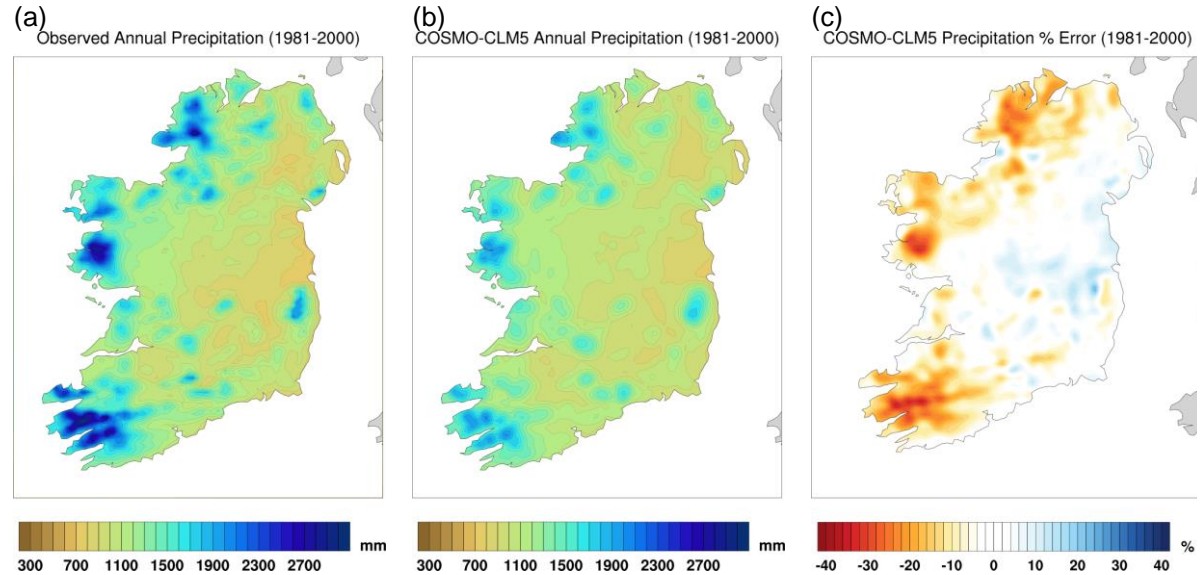


***Figure 3 Mean annual precipitation for 1981–2000. (a) Observations, (b) COSMO5-CLM-ERA-***
***Interim 4-km data and (c) COSMO5-CLM-ERA-Interim error (%).***


*Table 2 GCM and COSMO5-CLM Mean Absolute Error (%) uncertainty estimates through*
*comparison with gridded observations for the period 1976–2005. For each metric, the best- and*
*worst-performing scores are highlighted in green and red, respectively.*

| | 30-year average annual rainfall MAE % error | | |
|---|---|---|---|
| GCM | GCM Data | COSMO5-CLM-GCM 18 km | COSMO5-CLM-GCM 4 km |
| CNRM-CM5 | 16.5 | 14.1 | 11.8 |
| EC-Earth (r12i1p1) | 17.3 | 14.0 | 10.0 |
| HadGEM2-ES | 20.8 | 14.6 | 15.1 |
| MIROC5 | 26.0 | 18.2 | 15.6 |
| MPI-ESM-LR | 25.1 | 24.8 | 21.6 |

## Karst Groundwater Model
A semi-distributed pipe network model of the Gort lowlands has been developed by the
authors using urban drainage software (Infoworks ICM by Innovyze). This model simulates
both open channel and pressurised flow within the conduits with flooding on the land surface
represented by storage nodes with the same stage-volume properties of the physical turlough
basins (Morrissey et al., 2020). The model receives input from the four rivers as a time-varying
discharge which is computed separately using observed river gauging data provided by the
Office of Public Works (OPW) utilising established stage-discharge rating curves (Gill et al.,
2013a). Autogenic recharge across the catchment is represented within the model using sub-
catchments receiving a time-series of precipitation and evapotranspiration with inflows to the
pipe network controlled by a calibrated Groundwater Infiltration Module (GIM) within the
software. The downstream boundary condition for the model is the tidal level in Kinvara Bay
which is taken from Marine Institute observed data recorded at a buoy in Galway Bay. The
model was calibrated and validated over a 30-year period by matching the simulated
fluctuation of the groundwater-surface water interactions (i.e. turloughs levels) with observed
values and was found to represent the catchment with a very high degree of accuracy (Nash-
Sutcliffe Efficiency (NSE) & Kling-Gupta Efficiency (KGE) > 0.97). The full model setup and
calibration/validation process is presented in Morrissey et al. (2020).
The RCM rainfall and evapotranspiration data, described above, were then used to run the
groundwater flow model for each of the historical and future periods covering 24 simulation
periods in total (5 past & 19 future). Daily rainfall and evapotranspiration totals were output
from the RCM models in all cases and these values were used as input to Rainfall-Runoff (RR)
and karst models described below. When hourly totals were required to run the model, the
daily total was simply evenly distributed over the 24 hour period (this had no impact on the
model accuracy – see Morrissey et al. (2020) for further details).  The OPW have specified
the required allowances in flood parameters which should be made for planning purposes in
Ireland (OPW, 2019) for the "Mid-Range" and "High-End" Future Scenarios (MRFS & HEFS).
These provisions make allowances for both mean sea level rises and predicted land
movement of +0.55 m for the MRFS and +1.05 m for the HEFS. Therefore, to quantify the
combination effect of rising sea level with changing climatological conditions, the future
scenarios were also simulated with the tidal boundary condition adjusted to allow for predicted
increases in mean sea level at Kinvara Bay.
The karst model with uncertainty bounds as outlined in Morrissey et al. (2020) was used to
both simulate the past RCM period (1976 – 2005) and the future time slice 2071 – 2100. By
comparing the output from the RCM past and future simulations using the same calibrated
model the error or bias within the model itself is accounted for and the anomalies between
both periods represents the potential changes due to climate change. Other approaches for
climate change modelling with GCM's use bias correction techniques to correct the simulated
outputs for the past to correct the future and then utilise the differences between the two
corrected datasets. This process can introduce further error given that bias correction for such
models is an evolving field. The approach taken in this study has the advantage of eliminating
the need for bias correction (which is a recognised method in the literature) and accounts for
the karst model uncertainty.
# Results & Discussion
As outlined above, data from two time-horizons, 1976–2005 (the control) and 2071–2100,
were used for analysis of projected changes by the end of the 21st-century Irish climate. The
historical period was compared with the corresponding future period for all simulations within
the same group. This results in future changes for each model run; i.e. the difference between
the model future and past. While this strategy aims to remove the model bias, as outlined in
Nolan et al. (2017), a  level of uncertainty is common to all climate models which inherently
include bias particularly with respect to rainfall. Model uncertainty was compared to other karst
models to contextualise the results, the reported uncertainty of our model (3 -14%) is
comparable and within the same window when compared to other reported studies (e.g.
Mudarra et. al., 2019, Sofia et. al, 2020)
***Statistical analysis***
Considering that flood levels within turloughs are generally not normally distributed (Morrissey
et al., 2020), the non-parametric Kolmogorov–Smirnov statistical test was employed to test for
statistical significance of projected changes. The Kolmogorov–Smirnov null hypothesis states
that the past and future data are from the same continuous distribution. Small values of the
confidence level p cast doubt on the validity of the null hypothesis. The Kolmogorov–Smirnov
tests between each RCM past and future scenario show a high level of significance ($p \approx 0$),
meaning that the projected changes in the future flood level distributions are statistically
significant. For example, the projected changes in the Cumulative Distribution Functions
(CDF) for the MPI-ESM-LR RCM across the RCP2.6, RCP4.5 & RCP8.5 emission scenarios
at Coole Turlough are shown in Figure 4. A marked shift to the right is seen in the distribution
above flood levels (stage) of 5.5 meters above (Irish) Ordinance Datum (mOD), with the
RCP8.5 scenario showing the greatest shift with similar shifts in magnitude predicted for both
the low and medium emission scenarios. This indicates the likelihood of higher flood levels
being observed is higher in all future emission scenarios.

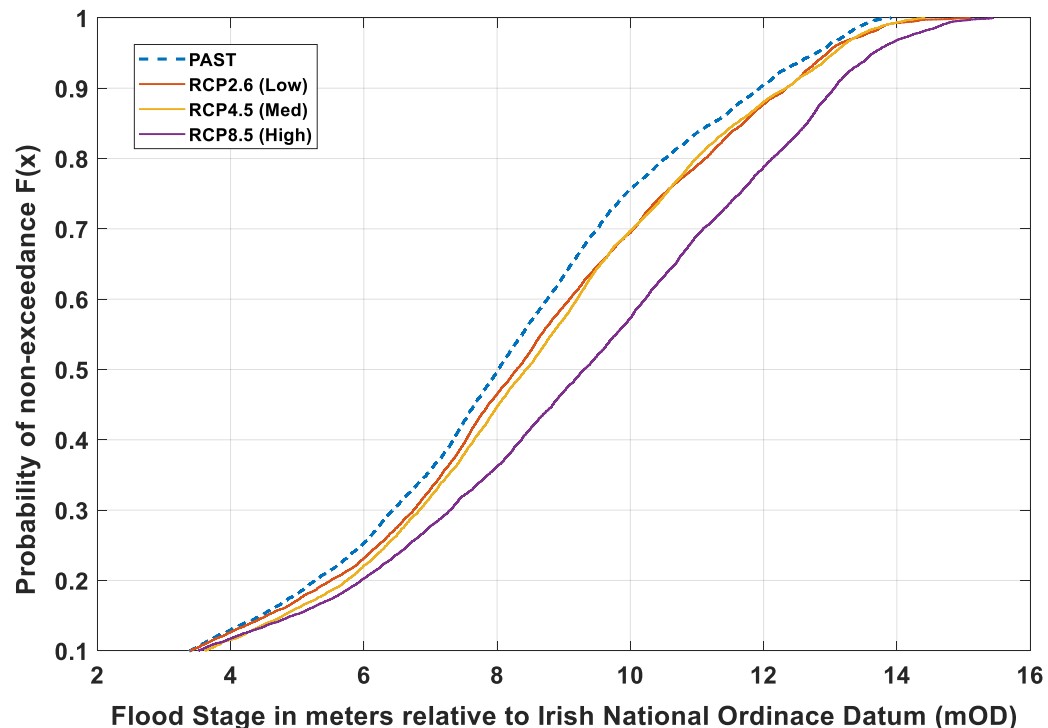


***Figure 4 Comparison of the non-parametric Cumulative Distribution Function (CDF) plots for the***
***past and future RCM scenarios using the MPI-ESM-LR GCM datasets at Coole Turlough [the y-***
***axis shows the probability F(x) of a particular flood stage (mOD) being less than or equal to x].***
***Note: Coole turlough is one of the key turloughs in the region and is representative of others***
***throughout the catchment.***

The predicted shifts in the data are further illustrated using box plots, as shown in Figure 5 for
Cahermore Turlough. In general, the RCMs predict progressively higher median and 75[th]
percentile flood levels with higher emission scenarios, with a few exceptions. The HADGEM2-
ES and MIROC5 RCM's predict similar future medians to the past, albeit with increased 75[th]
percentiles, whilst the MIROC5 results actually predict lower future 25[th] percentile flood levels.
Extreme values for all RCM future scenarios are increased with the exception of the RCP4.5
emission scenario for the MIROC5 RCM.  The reason for variation between various model
results is linked to the factors which impact karst flooding (e.g., which season, dry/wet event
impacts, winter vs summer, evapotranspiration vs precipitation, etc). The karst system
responds to previous cumulative rainfall along with existing flood level, so the pattern of rainfall
is crucial to the level and extent of flooding. Given that the GCM/RCM data are randomised,
the response of the karst model to the varying inputs will range. The use of ensembles
mitigates this potential area of uncertainty and gives a better indication of likely future
scenarios.

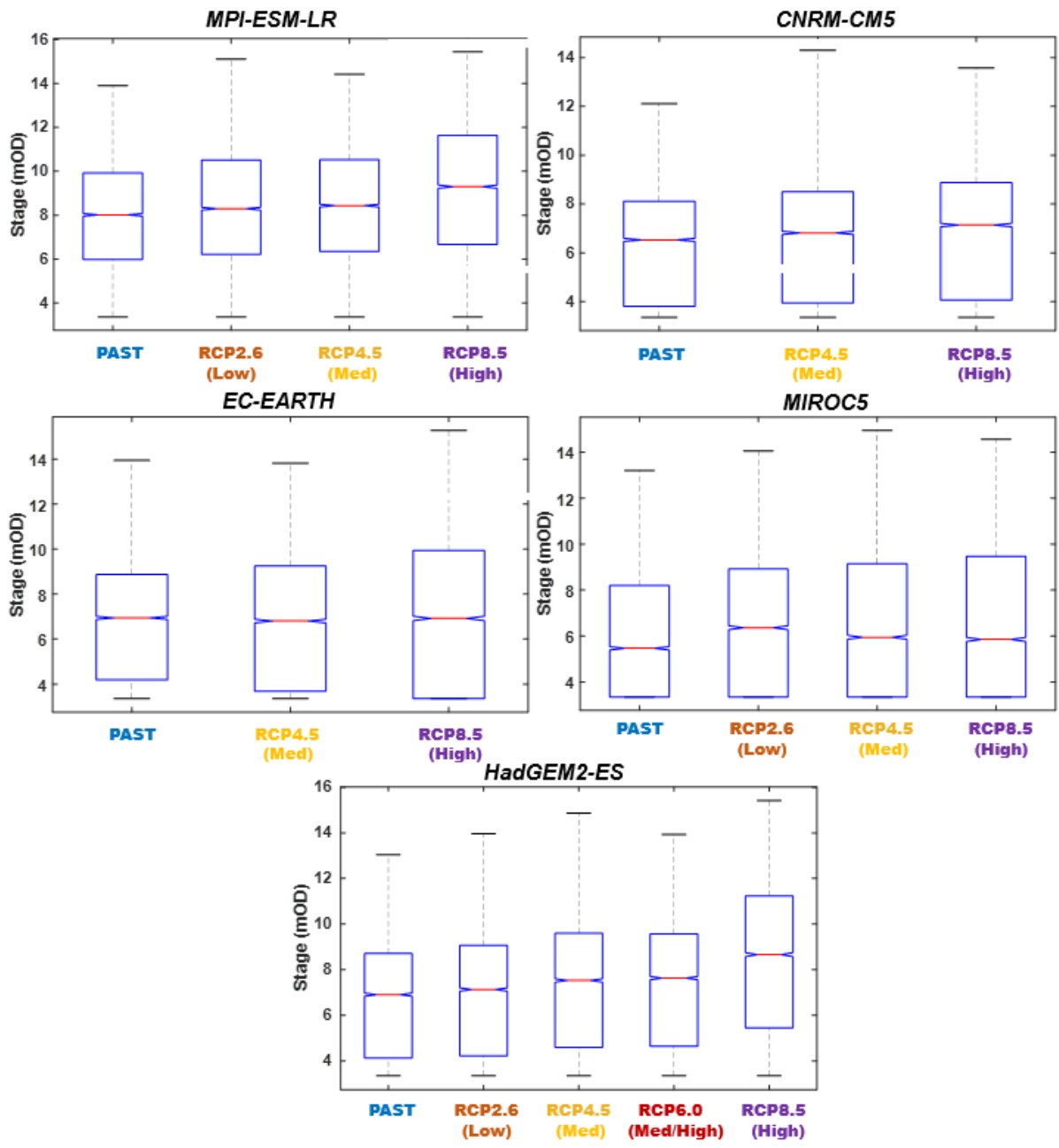


**Figure 5 Boxplots of model results for each of the RCM's showing past and future RCM scenarios at Cahermore Turlough. The central mark (red) indicates the median, and the bottom and top edges of the box indicate the 25th and 75th percentiles, respectively. Note: Cahermore turlough is one of the key turloughs in the catchment and is therefore representative of the general catchment trends.**


The Wilcoxon rank-sum test was employed to test for statistical significance of projected
changes in median flood levels. The Wilcoxon rank-sum tests the null hypothesis that the past
and future data are from continuous distributions with equal medians, against the alternative
that they are not. Each of the Wilcoxon rank-sum tests showed a high level of significance
($p \approx 0$) for the all ensemble scenarios across the entire catchment which therefore indicates
that the projected changes in the future flood level distributions and medians are statistically
significant.
*Implications for mean and recurrent flood levels and eco-hydrology*
In order to estimate the likely magnitude of change in future flood levels, an examination of
mean flood levels across the catchment was undertaken. Table 3 summarises the ensemble
average percentage change in sample means for all RCM scenarios across the catchment.
The models predict that ensemble mean flood levels will increase by an average 3.5% for the
low emission scenario and by 7.9% in the high emission scenario across the catchment.
Increases in mean water levels indicate either an increase in the magnitude of flood levels as
a whole, or an increase in the durations of flooding at higher elevations (or both). Further
analysis below reveals the nature of such mean flood level increases in more detail.

*Table 3: Ensemble average percentage change (%) in sample means for all RCM scenarios at all*
*groundwater flood nodes within the South Galway karst model domain (positive value indicates*
*increase in mean annual water level within the hydrological year)*

| Location within catchment | Ensemble Average % change in mean flood level | | | |
|---|---|---|---|---|
| | RCP2.6 | RCP4.5 | RCP6.0 | RCP8.5 |
| Ballinduff | 1.29 | 1.11 | 2.01 | 2.10 |
| Ballylea | 1.67 | 1.68 | 2.72 | 3.75 |
| Ballyloughaun | 0.14 | 0.21 | 0.18 | 0.60 |
| Blackrock | 3.83 | 4.12 | 6.30 | 8.98 |
| Caherglassaun | 8.14 | 8.29 | 12.20 | 17.62 |
| Cahermore | 5.61 | 7.01 | 9.75 | 15.42 |
| Castletown | 2.42 | 2.86 | 3.94 | 6.86 |
| Coole | 6.39 | 5.79 | 9.32 | 12.45 |
| Corker | 0.32 | 0.41 | 0.41 | 1.23 |
| Coy | 2.53 | 2.22 | 3.75 | 4.48 |
| Garyland | 7.32 | 7.72 | 11.78 | 16.48 |
| Hawkhill | 5.35 | 5.03 | 7.19 | 9.88 |
| Kiltartan | 1.25 | 1.44 | 1.86 | 3.80 |
| Mannagh | 0.82 | 0.87 | 1.51 | 1.94 |
| Newtown | 5.67 | 5.57 | 8.96 | 12.26 |
| **Catchment average** | **3.52** | **3.62** | **5.46** | **7.86** |

The impact of climate change on the seasonality of flooding in the turloughs was also
examined using the simulated climate data. The seasonality of flooding at turloughs typically
follows a pattern over the hydrological year (October – September) whereby flooding
commences in October/November with peak flood levels observed anywhere between
October and February. Figure 6 illustrates the ensemble shift in the seasonality of flooding
predicted to occur for the low, medium and high emission scenarios. The historical dataset
shows the peak frequency of flood levels generally occurring over the months December to
February. Each of the future RCM scenarios predict these frequencies will shift significantly
towards January and February and on into March for the high emission scenario. The
implications of peak flooding occurring later in the hydrological year (i.e. January / February)

are likely to mean flooding persisting later into late spring and even early summer as it usually takes a number of months for flood waters to drain down. This is especially significant for extreme flood events when a peak event occurring in late February could see flood water persisting until mid/late May. The associated impact for ecological habitats and indeed for farming (flooded lands adjacent to turloughs) in the catchment from this seasonal shift could be significant as persistent flooding could impact the growing season for wet grasslands and floral species. The impact of the timing of such peak events was demonstrated in the catchment during the two most recent extreme events. The extreme that occurred in 2009 peaked in late November and flood waters were largely abated by mid-March 2010, however flood waters from the extreme event of 2015/2016 which peaked in January 2016 persisted until late April 2016.

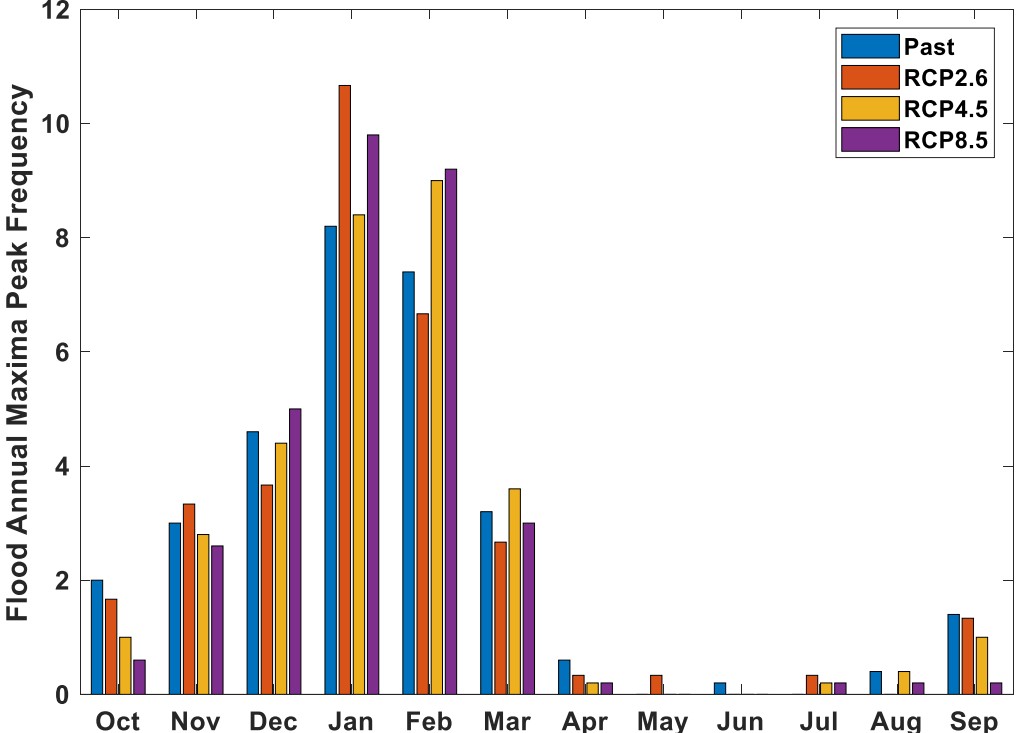

*Figure 6 Bar chart illustrating the seasonal shift in frequencies of peak annual flood levels at Coole Turlough over the hydrological year for all future RCM scenarios (with RCP 6.0 omitted). Note: Coole turlough is one of the key turloughs in the catchment and is therefore representative. The y-axis shows the number of times (or frequency) which the annual maximum peak flood level occurred during each month of the year by RCP ensemble.*

The spatial distribution of different vegetation communities in such wetlands is intimately entwined with the hydrological conditions (flood duration, flood depth, time of year of flood recession etc.), which change on a gradient moving up from the base of the turloughs. These ecohydrological relationships have been researched in multidisciplinary studies on these turloughs investigating links between the fluctuating hydrological regime and vegetation habitats, invertebrates, soil properties, land use and water quality (Kimberley et al., 2012; Irvine et al., 2018; Waldren et al., 2015) from which metrics have then be defined for the different key wetland habitats. For example, recent ecohydrological analysis the spatial distribution of vegetation habitats on four turloughs in this karst network (Blackrock, Coy, Garryland and Caherglassaun) over a 28 year period has revealed distinct differences between vegetation communities, from Eleocharis acicularis found at the base of the turlough

typically experiencing 6 to 7 months of inundation per year compared to the limestone
pavement community at the top fringes of the turloughs only flooded from 1 to 2 months per
year (see Figure 7). These differences in flood depth and duration are also reflected in a
gradient of times across the early growing season (spring) when the communities emerge
from the flood waters (and associated changes in air temperature and solar radiation).  Other
investigations on invertebrates in the turloughs (Porst and Irvine 2009, Porst et al., 2012) have
shown that hydroperiod (flood duration) has a significant effect on macroinvertebrate taxon
richness, with short hydroperiods supporting low faunal diversity. The study demonstrates how
different colonisation cycles occur in response to the seasonal hydrological disturbances (see
Figure 8).

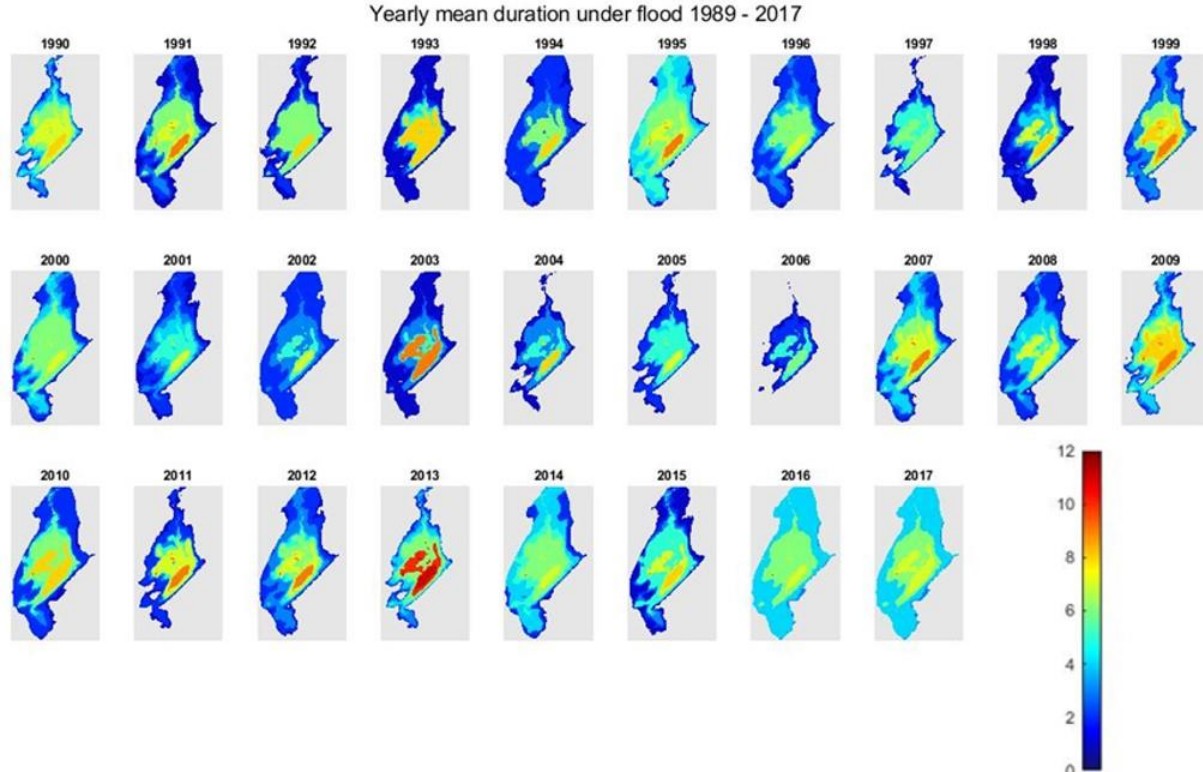

*Figure 7 Annual flood duration spatial profiles for Blackrock turlough over 28-year period.*

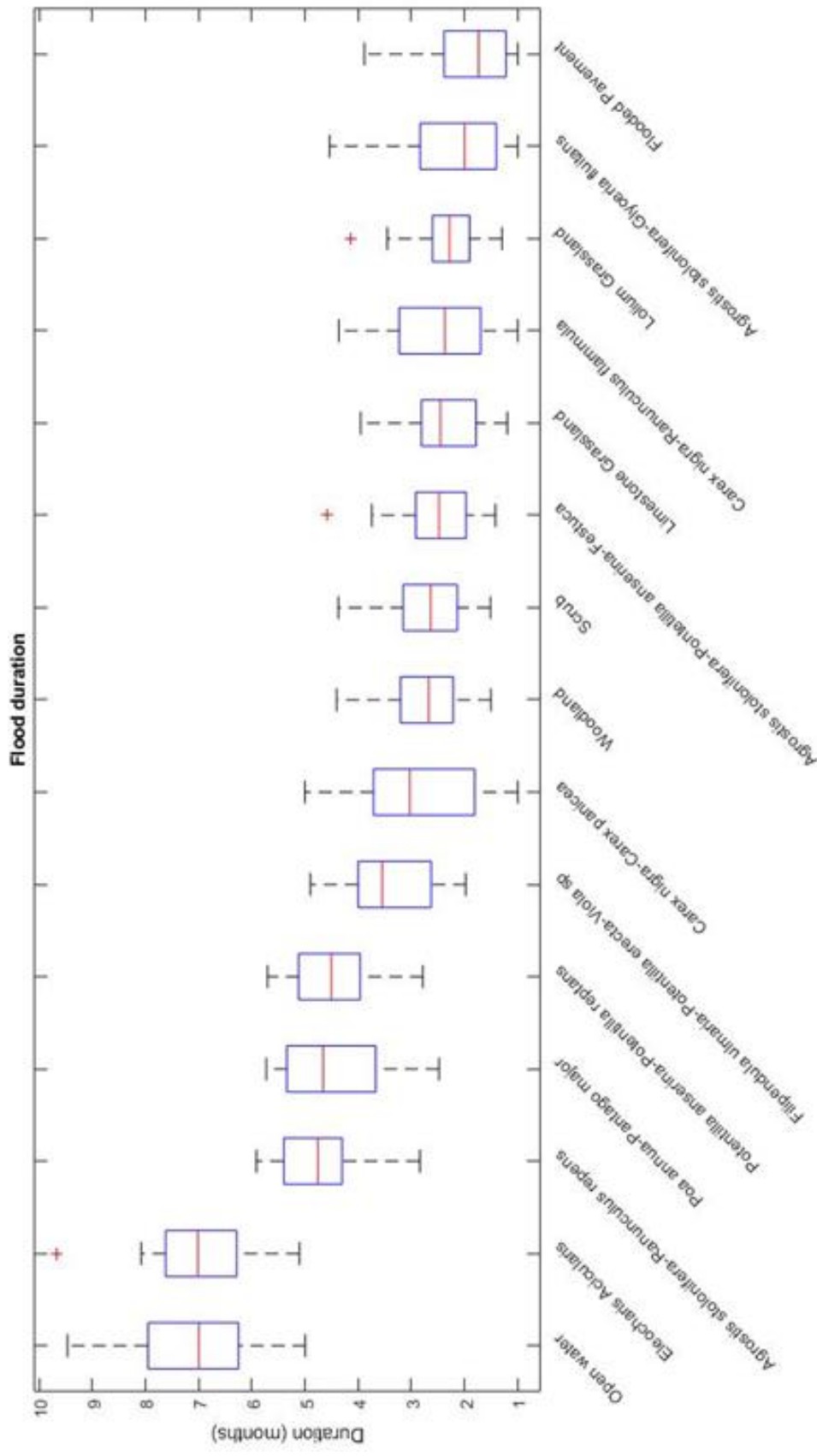


**Figure 8 The statistics of flood duration as a metric across the range of turlough vegetation communities averaged over four turloughs over a 28-yr period.**


The duration of inundation at various flood levels is of extreme importance, both from an ecological perspective in terms of wetland species distribution and survival and for extreme flooding in terms of the disruption to homes, transport links and agricultural land inundated by flood waters. An examination of the flood-duration curves across each of the five RCP scenarios (see Figure 9) indicates moderate to significant changes in the patterns of flood duration across the catchment. The MIROC5 RCM predicted the highest upward shift in flooded durations with a projected catchment average 99th percentile increase of 1015%. The EC-EARTH RCM predicts a reduction in low flood level durations and increase in high flood durations, with all other models generally predicting no significant shift in low to medium flood levels but upward shifts in flood durations at higher levels. Whilst the medium to low flood levels, which tend to be of more importance with respect to eco-hydrology, appear to be relatively unaffected, an examination of the more frequent flood inundation recurrences was undertaken using Annual Exceedance Probabilities (AEPs). The 50, 20 and 10% AEP flood levels were estimated for both the past and future scenarios using extreme value distributions. Given that the past and future horizons cover 30-year periods, it was possible to estimate the 10% AEP flood level with relative confidence. The annual maximum flood level series (using the hydrological year October to September) was extracted for each past and future scenario and an Extreme Value statistical distribution was fitted to the data. Each of the relevant flood levels were then estimated using the distributions and for each RCM the future and past values were compared to assess the projected future changes. The resultant ensemble catchment average changes in 50, 20 and 10% AEP flood levels across the various RCPs are shown in Table 4. The models predict a 4% increase in the 10% (10-year return period) AEP flood level for the low emission scenario and 10% increase in the high emission scenario. Similar increases are observed for the more frequent flood events indicating flooding of the turloughs will become more regular even at lower levels with the duration of dry or empty periods reduced. Given that the topography of each turlough basin varies widely (i.e. steep versus shallow sides), a 10% increase in lower flood levels will generally not be dramatic in terms of groundwater flooding, with respect to the risk to properties and/or damage and disruption throughout the catchment but will impact a large area as the side gradients tend to be shallow closer to the turlough bases. These changes in flood durations and the recurrence of flooding outside of the determined ecohydrological metric envelopes will undoubtedly have significant impacts for turlough eco-hydrology.

*Table 4: Ensemble catchment average percentage change (%) in 50,20 & 10% AEP flood levels for all RCM scenarios (positive value indicates increase in mean annual water level within the hydrological year)*

| RCM Scenario | Ensemble Average % Change in AEP Flood Level | | |
|---|---|---|---|
| | 50% AEP | 20% AEP | 10% AEP |
| RCP2.6 | 2.92 | 3.88 | 4.25 |
| RCP4.5 | 4.52 | 5.63 | 6.05 |
| RCP6.0 | 4.67 | 4.60 | 4.58 |
| RCP8.5 | 8.97 | 9.76 | 10.07 |

When assessing the impacts of climate on groundwater flooding in the lowland karst of Ireland, the extreme values within the data are of most interest. Given that the future horizon considered for all scenarios covers the 30-year period between 2071 – 2100, this is not a long enough period from which to estimate the 1% AEP with any degree of certainty. In addition, due to the non-parametric nature of the data, it was not possible to employ the use of extreme value statistical distribution to estimate values without introducing large margins of error. For example, the peak values between the past and future scenarios were found to vary between -1.6% and +16.5% across each of the various future RCM scenarios; however, there is no

statistical test to determine if these changes are indicative of a trend or linked to random
chance within a 100-year future time interval. Trends in the 95th and 99th percentile time-series
values have previously been used successfully to test for statistically significant trends in
extreme values in climate change analysis (Franzke, 2013). In order to establish if a
statistically significant difference existed in the future RCM scenarios, the Kolmogorov-
Smirnov two sample test was therefore used with all values below the 95th percentile excluded.
***Implications for extreme flood events***

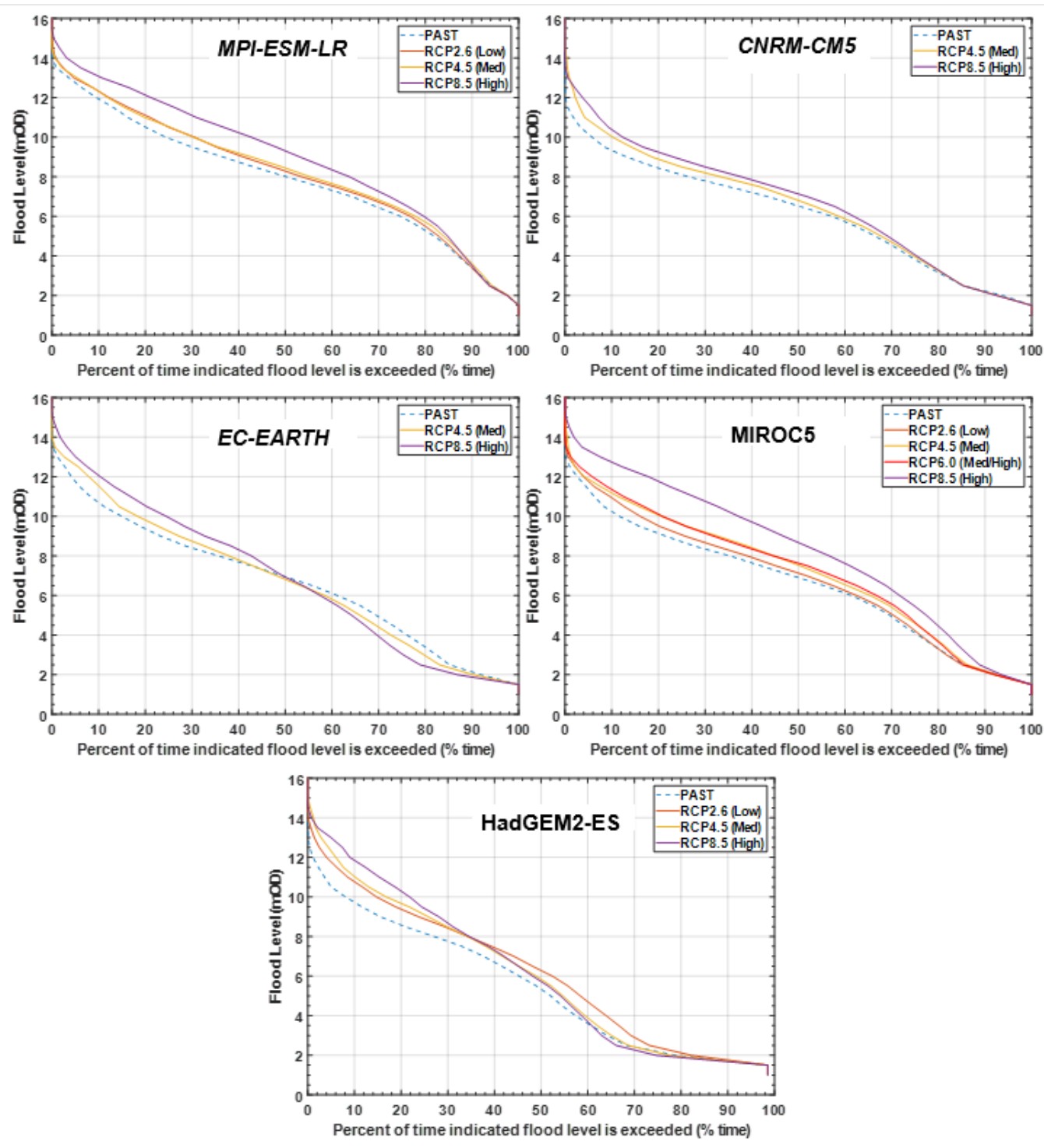

52ₒ
***Figure 9 Flooded duration curves at Coole Turlough for each of the RCM scenarios***
The null hypothesis was rejected for all future RCM scenarios indicating that the differences
between the distributions in the upper (and most extreme) range are statistically significant.
Sample CDF plots of past and future scenarios for the MPI-ESM-LR RCM at Coole Turlough
utilising data values above the 95th percentile are given in Figure 10.

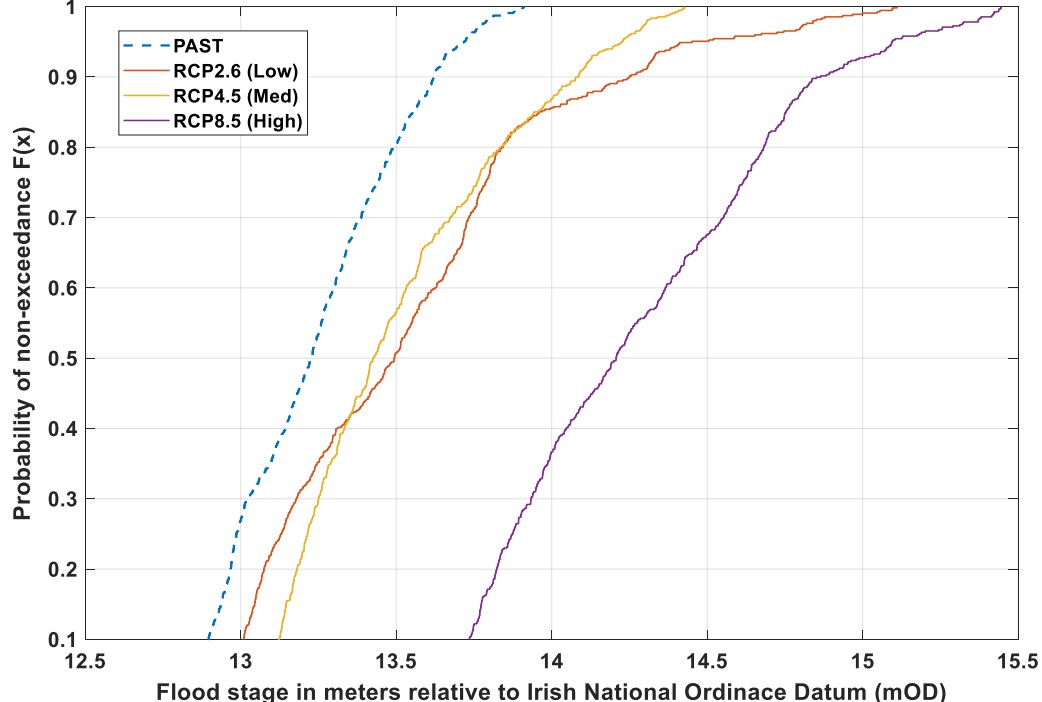


*Figure 10: Comparison of the non-parametric Cumulative Distribution Function (CDF) plots for*
*the past and future RCM emission scenarios using the MPI-ESM-LR RCM datasets at Coole*
*Turlough with values below the 95th percentile excluded (annual maxima levels)*

Given this test indicates that a future trend exists, the 95th and 99th percentile values at each
model node were then calculated for each of the ensemble RCM simulations and the
ensemble average percentage change between each of the past and future sceneries was
used to determine the ensemble average across the entire catchment (see Table 5). All future
scenarios predict an increase in the 95% percentile flood level across each model node with
the catchment average ranging between +3.8% (future-low) and +10.3% (future-high). It must
be noted that two of the turloughs in the catchment (Ballinduff and Coy) show very little change
in 95th percentile values across all future scenarios. Both of these turloughs are almost always
permanently flooded with Ballinduff having a relatively narrow range of annual fluctuation in
flood levels (<4 m). Both locations flood to their notional maximum level far more frequently
with further increases in flood water levels controlled by either overland flow paths or sinkholes
at higher elevations. This is not representative of the majority of other flood locations within
the catchment, which reach their notional maximum flood levels far less frequently. Hence, it
should be noted that removing these two turloughs from this analysis would only serve to
further increase the catchment average values shown in Table 5.

*Table 5: Ensemble percentage change (%) in 95th percentile flood levels for all RCM scenarios*
*at all groundwater flood nodes within the South Galway karst model domain (positive value*
*indicates increase in 95th percentile water level within the hydrological year)*

| Location within catchment | Ensemble Average | | | |
|---|---|---|---|---|
| | RCP2.6 | RCP4.5 | RCP6.0 | RCP8.5 |
| Ballinduff | 0.05 | 0.06 | 0.06 | 0.11 |
| Ballylea | 2.19 | 2.63 | 3.43 | 7.97 |
| Ballyloughaun | 0.51 | 1.74 | 1.53 | 4.78 |
| Blackrock | 3.87 | 4.93 | 5.73 | 10.51 |
| Caherglassaun | 5.84 | 6.99 | 6.88 | 17.09 |
| Cahermore | 5.84 | 7.47 | 7.14 | 16.65 |
| Castletown | 5.65 | 7.76 | 7.73 | 14.31 |
| Coole | 5.74 | 7.87 | 7.67 | 14.80 |
| Corker | 3.27 | 3.57 | 6.27 | 7.56 |
| Coy | 0.31 | 0.73 | 0.38 | 0.89 |
| Garyland | 5.74 | 7.41 | 7.60 | 15.03 |
| Hawkhill | 5.74 | 7.88 | 7.67 | 14.80 |
| Kiltartan | 5.32 | 6.33 | 6.08 | 11.33 |
| Mannagh | 1.25 | 2.24 | 2.59 | 3.66 |
| Newtown | 5.74 | 7.50 | 7.67 | 14.80 |
| **Catchment average** | **3.80** | **5.01** | **5.23** | **10.29** |

A further calculation was then undertaken which estimated the percent change in the
frequency of days with peak flood levels greater than the current 95th and 99th percentiles,
respectively. The simulations project 64 to 205% increases for the 95th percentiles across the
RCM scenarios with 171 to 621% increases in 99th percentile exceedance frequencies. That
is, flood levels that are currently considered unusually high will become much more common.
Given that mean flood levels across the catchment were also shown to increase by between
3.5 to 7.9%, it follows that an upward shift in the more extreme flood levels (i.e. 1% AEP) will
also occur. Whilst this analysis indicates that an increase in 1% AEP flood levels across the
catchment will likely occur, the magnitude of the increase will be controlled by the natural
overland spill points between the turloughs and also the capacity of potential linked overland
flow paths to the sea.
The spatial extent of the 1% AEP flood for the study catchment was carried out and compared
to a similar map produced for the same flood using the RCP4.5 (Med) ensemble results – see
Figure 11. The 1% AEP flood predicts that 24.18km$^2$ will be flooded during the peak. This
compares to 29.77km$^2$ inundated during the RCP4.5 (Med) scenario (a 23% increase). |It must
be noted that Figure 11 only includes the food extents of the subject model and flooding from
other sources (not simulated) would also likely occur during such an event.

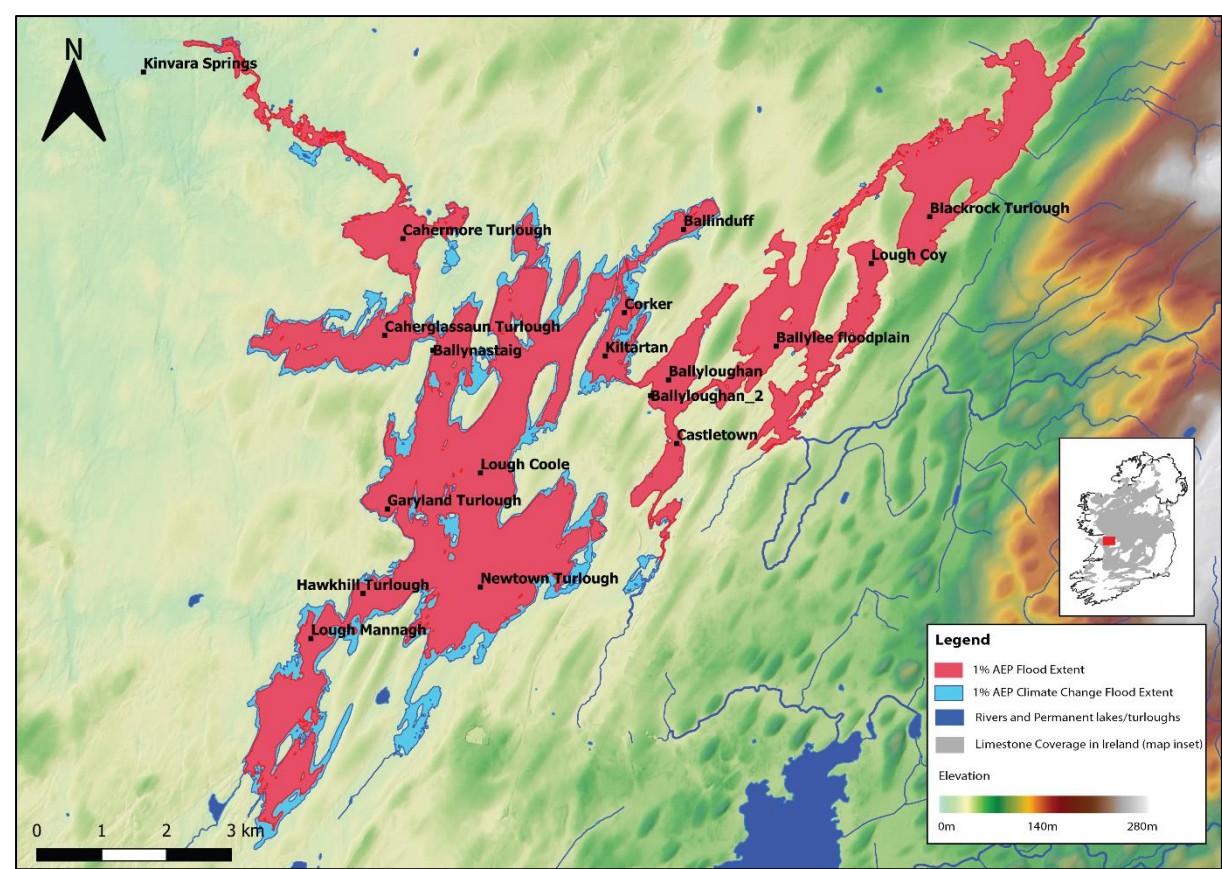


**Figure 11 Comparison of the spatial extent of the 1% AEP flood event for the study catchment and the associated increases predicted during the RCP4.5 (Med) ensemble scenario.**



### *Impact of rising mean tide levels*

All 19 future RCM scenarios were re-simulated with the downstream tidal boundary condition increased to reflect projected rises in mean sea level. The tidal boundary signals used in the future RCM scenarios were therefore shifted upwards by 0.55 m and 1.05 m respectively and all future scenarios were re-assessed. No statistically significant change in any of the resulting distributions was found however, when compared to the future RCM scenarios with no sea level increases. This indicates that the differences between the distributions with mean sea level increases are statistically insignificant and that rises in mean sea levels of up to 1.05 m will have little impact in this karst catchment over and above the impacts of changing climate. Similarly, there was no appreciable change in average or 95th percentile flood levels across the catchment (<0.05 m). Minor changes in peak levels (<3%) were observed at Caherglassaun turlough which is the closest to the sea and where a tidal signal is observed at low flood stages; this minor change however, was not observed at any other location. The observed changes at Caherglassaun were not enough to reject the null hypothesis for any statistical test. An examination of the pattern of outflows from the system at the springs at Kinavara confirms that these results are to be expected. The majority of outflow from the system (through the intertidal springs) occurs during the ebb tide when the bay is essentially empty (elevation <-2.5 mOD) or emptying. Even a mean sea level rise of 1.05 m would only increase the bottom elevation of the ebb tide to approximately -1.5 mOD which would still allow equivalent volumes of water to drain from the system during ebb tide. In addition, an examination of the spring outflows for the historical and future RCP scenarios through the

ebb/flood tidal cycle showed water was still flowing out of the system as the tide rises due to
the pressure head between groundwater in the aquifer (and the turloughs) and the springs.
A comparison was made between the finding sof this study and other karst studies which
considered climate change. A study undertaken by Nerantzaki & Nikolaidis (2020) which was
similar in nature (i.e. use of GCM and RCM data with karst models) and indicated that a
reduction of karst spring flow of between 14 - 25% could occur under climate change scenarios
(authors used a blended rainfall spectrum from RCP2.6 and RCP8.5). This range is
comparable to some of the results observed in this study. Similarly, other studies focused on
the impacts of karst aquifer due to climate change utilise GCM/RCM and various emissions
scenarios (Pardo-Igúzquiza et al., 2019) but are concerned with impacts to recharge (and
spring water availability) and flooding/eco-hydrology are not considered. It is therefore difficult
to provide direct comparisons with this current study; however the authors are confident the
projections reported in this study are broadly in line with other international studies.

**Conclusions**
**Groundwater Flooding**
It has been established that the long-term trends of the lowland karst aquifer dynamics (e.g.,
spring discharge, groundwater levels and groundwater flooding) are affected by precipitation
patterns (intensity & accumulation) over preceding weeks and months leading up to peak
water levels (peak flood events) typically late in the winter or early spring (Naughton et al.,
2012). Quantifying the impact of changing rainfall patterns is therefore of upmost importance
when considering future groundwater flood risk in such lowland karst catchments. Whilst
significant variations in the magnitudes of predicted future increases in flood levels were
observed in this study, the underlying trend in the RCM data simulated is predicting increases
in mean annual flood levels (groundwater levels), 95[th] and 99[th] percentile levels and most
significantly in flood durations particularly at higher (and more extreme) flood levels. This study
has demonstrated how the spatial extent of the 1% AEP flood will expand which is useful for
flood risk mapping purposes. Each of the various downscaled GCM datasets predicted
statistically significant increases in all relevant flooding statistics and notably a shift in the
seasonality of the flooding. This shift will likely compound the impact in the catchment given
that the existing summer "dry" period may be curtailed. The projected large increases in the
frequencies of the existing (past) 99[th] percentile exceedances of up to 1015% clearly
demonstrate that what is currently considered to be high or extreme flooding will become more
of a regular occurrence in the future. In terms of planning for future development or indeed
developing flood alleviation projects for such lowland karst systems, being able to predict the
projected changes in mean flood levels and extreme events will be vital in order to ensure that
developments proceed with minimal risk to property or human life. In this study catchment this
could result in potential flood alleviation channels being sized to accommodate considerable
larger flows that what may be considered sufficient based on current conditions. The
implications of this study for similar karst catchments and climate zones with high recharge
rates and significant seasonal variations in groundwater levels are equally significant and
could also impact on other activities such as tunnelling and mining in such karst environments.

**Eco-hydrology**
Ecosystems which rely on groundwater to sustain wetland conditions are at particular risk to
changes in inundation fluctuation regimes brought about by climate change. This study has
shown that the pattern of flooding at turloughs in the west of Ireland is likely to change
significantly with higher mean flood levels over longer durations. Different unique habitats have
developed under such cyclical envelopes of hydrological conditions, presenting a spatial
gradient of different communities that can exist under the different conditions moving up from
the base of the turlough. Hence, the results of this climate change study predict that a change
in the hydrological regime is likely to cause associated changes in the location and extent of
these habitat zones within turloughs. Furthermore, some of these habitats may be at threat
due to the predicted shift in the seasonality of flooding to later in the hydrological year, causing
a delay in the critical early growing season for wetland grasses and flora. Ongoing studies
have been investigating the differences in prevailing air temperature and solar radiation for
the vegetation communities across the turloughs as they come out of the winter flood regime
at different times and are first exposed to air in the spring. The increase in more extreme
events could also have a detrimental impact to fringing habitats which develop along the
perimeter of these sites (typically woody shrubs and trees or limestone pavement
communities) which would be severely impacted were they to become flooded on a more
regular basis. An argument could be made that the habitat zones could simply be shifted
upwards in elevation, essentially expanding the extents of the wetlands. However, given that
turloughs are often located within defined basins, the room for their "growth" is constrained
and the loss of some habitat is likely to be unavoidable. For other similar groundwater
dependent ecosystems in similar climate zones in karst such as fens the implications of
fluctuations in future groundwater levels and flows are equally significant.
In the wider context, this study has shown that the use of complex transient groundwater
models with the output from RCM models can provide specific and targeted information on the
likely effects of climate change on groundwater levels, flooding and eco-hydrology. More
specifically this methodology can clearly be transferred to study other karst based GWDTEs
such as calcareous fens and poljes.

674                                    **Acknowledgements**

This work was carried out as part of the scientific project "GWFlood: Groundwater Flood
Monitoring, Modelling and Mapping", funded by Geological Survey Ireland and by Galway
County Council. The work also represents outputs from research funded by the Office of Public
Works and the Irish Research Council. The authors would like to thank the Irish Meteorological
Service (Met Eireann) for the provision of rainfall data, Galway County Council for the provision
of aerial photography and GIS data, and the Office of Public Works for the provision of LIDAR,
hydrometric and aerial photography data. The authors would also like to thank two anonymous
reviewers whose input and suggestions have added greatly to this paper.

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
