# Peer review of "IMPACTS OF CLIMATE CHANGE ON GROUNDWATER FLOODING AND ECOHYDROLOGY IN LOWLAND KARST"

_Hydrology and Earth System Sciences, 2020_

## Referee Comment (RC1) · Anonymous Referee #1 · 9 Aug 2020

In the submitted manuscript, Morrissey et al use a semi-distributed karst model to estimate the possible impacts of climate change on a lowland karst system in Ireland both in terms of groundwater flooding and ecohydrology. Their model predicts that groundwater events that are currently considered extreme will increase in the future. In addition, a future shift of flood seasonality is simulated by the model, which the authors suggest will likely affect ecological systems.

Generally, the study is well-written and concise. Using a well-established karst model to estimate future groundwater flood frequency and ecohydrological implications is novel and of high interest for the readers of Hydrology and Earth System Sciences. As

elaborated in the technical/specific remarks below,

(1) comparing the RCMs' control periods simulations with meteorological observations may help to assess their trustworthiness.

(2) estimating the uncertainty of the hydrological predictions may help avoiding wrong conclusions about moderate changes that are smaller than the prediction uncertainty

(3) A comparison with projected changes of karst processes of other karst model applications to assess future changes may permit some evaluation of the predicted changes of this study.

Overall, I am confident that the authors will be able addressing these point within the frame of minor revisions.

Abstract:

Some more information about the methods (model setup, climate scenarios) is necessary.

Introduction:

Line 51: Typo "changeon"

There is some work on groundwater level frequencies and climate change by Bloomfield & colleagues and some semi-distributed modeling of GW levels and climate change by Brenner et al (2018, NHESS), which may be useful for the state-of-the-art

Regional Climate Modelling

This section is a short of a general review. Please either add study site specific information (which RCMs are available? how have they been established?) or move to Introduction

Methodology

Is it possible to compare the model ensembles of the historic/control time period with

observed climate data?

Lines 192-193: "GCM to 18 km to 4 km" Please rephrase into full sentence.

Table 1: Not all RCPs are available for all of the models. How was this handled when working with the model ensembles?

Figure 2: numbers in grey circles within figures not elaborated in caption

Karst Groundwater Model:

Since the model parameters are estimated via calibration, it would make sense providing some estimate about their uncertainty especially since the model is used for prediction far in the future. Moderate predicted changes in flood frequency might be still within the envelop of the simulation uncertainty.

Results & Discussion:

A general assessment on how much you trust the model projections is necessary. A comparison of the karst hydrological projections of this model with the hydrological projections of other karst models might be useful for that. Which processes are prediction to become more pronounced by this model and is this in agreement to the projected changes in hydrological behavior of other karst simulation models that were run with climate projections? Do the projected karst hydrological changes agree with the conceptual understanding of the lowland karst?

The discussion about ecohydrology could be a bit more specific. Which negative specific consequences may occur? Which plants/animals are affected most? Maybe one or two specific examples would help.

Figure 4: please enlarge x-/y-axis label font sizes

Line 325: Typo "Figure 5illuminates"
* * *
203, 2020.

---

## Referee Comment (RC2) · Anonymous Referee #2 · 15 Oct 2020

Questions to address:

1. Does the paper address relevant scientific questions within the scope of HESS? Yes. What are the impacts of climate change on flooding in lowland karst areas?

2. Does the paper present novel concepts, ideas, tools, or data? Yes. Existing modeling tools and datasets are integrated and analyzed in a novel way.

3. Are substantial conclusions reached? Yes. Flooding is likely to increase significantly (both in magnitude and duration) in the studied catchments due to climate-induced changes in precipitation patterns. Sea level rise is not likely a factor in increasing flood risk in the study catchments.

4. Are the scientific methods and assumptions valid and clearly outlined? Yes. An ensemble of regional climate models was used to generate inputs to a pipe-flow model of groundwater flow in the study catchment. Thorough statistical analyses demonstrate that the model results are significant.

5. Are the results sufficient to support the interpretations and conclusions? Yes.

6. Is the description of experiments and calculations sufficiently complete and precise to allow their reproduction by fellow scientists (traceability of results)? Yes.

7. Do the authors give proper credit to related work and clearly indicate their own new/original contribution? Yes.

8. Does the title clearly reflect the contents of the paper? Somewhat. This study does an excellent job of thoroughly analyzing the potential changes in flooding patterns in lowland karst areas due to climate change, but there is no in-depth analysis of the impacts on the ecohydrology. The one paragraph discussing ecohydrology at the end of the paper includes no citations, data, or analysis. I would therefore recommend changing the title to "Impacts of climate change on groundwater flooding in lowland karst".

9. Does the abstract provide a concise and complete summary? No. There is a misleading emphasis on ecology. The paper focuses on flooding, therefore, the abstract should as well. A sentence should be added indicating why flooding is a concern. The first sentence is also misleading – this paper is focused on turloughs, and the abstract should therefore reflect that by beginning with a clear one-sentence description of turloughs. However, the rest of the abstract is quite good.

10. Is the overall presentation well structured and clear? Mostly. The motivation for studying flooding should be presented at the beginning – the authors indicate that flooding is a concern but do not explain why until the second-to-last section. I would recommend moving the bulk of the description of why flooding is harmful to the intro-
duction, possibly under the sub-heading "Motivation". Also, the introduction emphasizes drought and ecological impacts, but drought is not discussed at all in the rest of the paper, and ecological impacts are discussed only briefly.

11. Is the language fluent and precise? Somewhat. The manuscript would benefit from a thorough reading for grammar, typos, and consistency. There are unnecessarily awkward third-person sentence constructions.

12. Are mathematical formulae, symbols, abbreviations, and units correctly defined and used? Mostly. Several abbreviations and units are used without being first defined. Abbreviations are not always consistent.

13. Should any parts of the paper (text, formulae, figures, tables) be clarified, reduced, combined, or eliminated? The section on ecohydrology is lacking references and specificity, and should therefore either be expanded into a full, well-referenced discussion or eliminated.

14. Are the number and quality of references appropriate? Yes (except for abovementioned exception in the ecohydrology section).

15. Is the amount and quality of supplementary material appropriate? Not applicable.

General comments:

The authors present a well-thought-out modeling study with methods that may be applicable to other lowland karst catchments vulnerable to climate change impacts, and with findings that are likely to be of great interest to planners in responding to climaterelated stresses such as flooding. The article is generally well-organized and clear, but requires a thorough grammar/typo/consistency revision. My only substantive critique is that the ecohydrology section is quite thin and should either be overhauled or eliminated. Finally, the paper would be more broadly relevant and interesting if it also incorporated a spatial analysis of flooding (which may or may not be possible given the modeling setup). However, if it is possible, I believe it would be well worth the authors' HESSD
time to expand the scope of the analysis slightly to include this (see more detailed comments below).

Specific comments: individual scientific questions/issues

1. Introduction:

a. Line 45: Please describe the projected shifts in precipitation patterns – Increase/decrease? Change in seasonality? Change in spatial distribution? Intensity? Frequency?

b. Line 58 & 68: The terms habitat and ecosystem are used interchangeably throughout the text. Usually habitat describes conditions appropriate for a specific organism or type of organism of interest, while ecosystem describes the entire biotic and abiotic community. Unless the authors have a specific organism/species in mind that is at risk or of particular local significance, it is best to use the term "ecosystem" rather than "habitat". Also, it is not clear here what an "eco-hydrological habitat" means. Is this referring to a groundwater-dependent ecosystem? If so, it is best to define and use the latter term consistently (it is introduced as an abbreviation on line 68 but then not used again). If not, please define "eco-hydrological habitat" and specify the organism for which this habitat is present. Or just consistently use the term "groundwater-fed wetland" if that is the particular ecosystem present in the study area.

c. Line 68: Please include a sentence explaining why droughts and floods threaten groundwater-dependent ecosystems.

d. Line 71: Please include a sentence or clause explaining why karst models are more difficult to couple with GCMs/RCMs.

e. Line 76: Please briefly define groundwater flooding vs. fluvial flooding.

f. Line 82: Please include at least a sentence explaining what types of damage and disruption are caused by flooding – infrastructure damage, cutting off transportation access, destruction of homes, preventing planting or harvesting of crops? Please ex-

**HESSD**
pand on how is it different from fluvial flooding. Please also briefly explain the potential impacts of drought on human society (infrastructure, agriculture, water supply, etc.).

g. Line 83: This section is a bit scattered. It would benefit from being restructured. Either discuss the effects of both flooding and drought on groundwater-dependent ecosystems, and then discuss the effects of both flooding and drought on human society, or have a paragraph on drought and a paragraph on flooding, with human and ecosystem impacts of each. I would suggest focusing on flooding and human impacts, since the ecosystem impacts discussion later in the paper is not as well fleshed out and there is no substantial discussion later of drought.

- 2. Study Catchment
- a. Figure 1:

i. Please either include topography underlay or indicate flow directions – it is not clear what the predominant flow patterns are from the map as is. ii. Please include an inset showing the location of the study catchment within the country, ideally with major karst areas in the country indicated (maybe use the World Karst Map freely available GIS data?).

iii. Please label all locations referenced in the text (Kinvara Bay, Galway Bay, Gort Lowlands, Galway Lowlands, Slieve Aughty Mountains, etc.).

iv. Does the study area have a name? If so, include it.

v. The caption says that model nodes are represented, but they are not clearly indicated or visible on the map?

b. Line 100 & 120: Though I am not very familiar with turloughs, my understanding is that they are a type of polje. This should be clearly stated when turloughs are defined and introduced, so that readers familiar with karst generally but not Irish karst specifically will be able to place these features in the context of other karst systems. The definition of turloughs is currently split between lines 100 and 120 and should be
condensed into one section. Also, the current wording at line 120 makes it seem as though the term turlough refers only to the lake (when present), while line 100 makes it seem as though it refers to the depression even when dry. Please clarify.

c. Line 122-126: Please specify what type of damage was caused by flooding (see earlier comment).

- 3. Regional Climate Modelling
- a. This is a very nice description of model downscaling!
- 4. Climate Models and Methods

a. It would be helpful to have a table or bullet points giving a brief summary of the features, strengths, and weaknesses of the five global datasets being used. Possibly this could be included in Table 1.

b. Please give a brief explanation of the RCPs – what does the number attached mean (CO2 concentration?), and what does it represent (low, medium, high emissions scenario?). Throughout the rest of the paper, please use consistent terminology and color schemes for the RCPs. The text's readability would be improved if, once the different RCPs were introduced, they were then consistently referred to as low, medium, and high emission scenarios, and labelled as such in the figures and tables. As it is, they are alternately referred to by a confusing range of abbreviations and descriptive phrases, and are represented by different colors in each figure. The clearest visual representation would be a sequential color gradient from low to high emissions. Again, possibly the definitions could be included in Table 1.

c. Figure 2: This is a nice visualization of the climate models.

i. Are these the means of all 5 GCMs for each RCP? Please clarify which scenarios are being displayed.

ii. Please label the colorbar more specifically - percent increase or decrease from
observed mean historical precipitation? Or is it from the mean modeled precipitation across the modeled past period (1975-2005)? Not clear.

iii. Correct the RCP labels to include the decimal point. Adding high/medium/low labels like in Fig. 3 would be helpful. See previous comment about consistent naming.

iv. Please define/explain the small numbers in gray bubbles in the caption.

v. Please include the outline or point location of the study catchment within the larger maps of Ireland.

5. Karst Groundwater Model

a. Line 228: Please explicitly state who developed the model. The current third-person passive construction muddles authorship.

- 6. Results & Discussion
- a. Line 266: Please specify the direction/type of bias (overpredict/underpredict? Etc.)
- 7. Statistical analysis

a. Line 274: What are typical ranges of p values and what values would indicate statistical insignificance?

b. Figure 3: Please label axes more clearly. Y-axis: Probability of non-exceedance F(x). X-axis: Spell out whatever mOD is an abbreviation for. In caption, state primary takeaway from figure: Probability of non-exceedance is lower for future climate scenarios compared to past, therefore flooding is more likely in all future scenarios. Why was Coole Turlough chosen? Is it representative of other turloughs in the study area?

c. L 292: Please include a brief discussion of possible reasons why the HADGEM2-ES and MIROC5 datasets might predict little to negative change in flood levels.

d. Figure 4: Why was Cahermore Turlough chosen? Is it representative of other turloughs in the study area? Why do Fig. 3 & 4 represent different turloughs? See
previous comments about labeling and coloring of RCPs.

8. Implications for mean & recurrent flood levels & eco-hydrology

a. Table 2: For planning purposes, it would be useful to know what the mean flood stage is for each location and scenario as well.

b. Line 335: Please discuss the effects of late-season flooding in more detail. Roughly how much farmland is in the turlough-adjacent flood zone? Are there any studies of flood impacts on wet grasslands and the general ecology of these systems? Also, please define "knock-on effect" or use a more widely understood phrase.

c. Figure 5: This is a nice figure. It would read more clearly if the colors and labels for the RCPs matched other figures (see previous comment). Please explain why Coole Turlough was chosen – is it representative of the others?

d. Line 346: This sentence or something like it should be included in the abstract and in the introduction, to explain why flooding is a concern. Even better would be to provide estimates of how much agricultural land, how many residences, and how much major infrastructure is in the affected area.

e. Line 376: Please provide some sort of evidence for this claim.

f. Figure 6: It is hard to tell apart RCP 4.5 and 8.5 because the colors are so similar. See comments about consistent color use across figures.

9. Implications for extreme flood events

a. Figure 7: See comments for Figure 3.

b. Line 415: Be cautious of stating that something is definitively proven, especially statistically. The K-S test indicates that the results are statistically significant. These are two different things.

10. Impact of rising mean tide levels
a. This is interesting! Would you expect areas with small to no tidal fluctuations to see more of an impact from rising sea levels?

**11. Groundwater flooding**

a. For planning purposes, it would be particularly interesting to see predictions of the spatial distribution of flooding. Is it possible to include some analysis and maps of the spatial extent of peak flooding under different scenarios? What about maps of the catchment showing number of flooded days per year at each location? Or the last day of spring flooding? If the modeling approach presented in this paper could generate such maps for this and other catchments, it would be a powerful adaptation planning tool, and I think it would be well worth the time to add these analyses.

**12. Eco-hydrology**

a. This section does not cite any references to support the claims made. There are several interesting and valuable ideas, but they are not discussed in much depth, nor are they supported by evidence. I would therefore recommend either removing this section and the discussion of flooding impacts on the turlough ecosystems entirely, or taking the time to develop it properly (the latter option would be an excellent contribution and I hope that the authors will choose to explore this in more depth).

b. See previous comments about terminology with respect to "habitat", "ecosystem", "groundwater-dependent ecosystem", etc.

c. There is no discussion of drought in the conclusions. Drought should therefore either not be mentioned in the introduction, or it should be made clear that drought is not within the scope of this paper.

d. What is the potential transferability of this approach to other locations? This would be worth discussing briefly somewhere in the conclusion.

Technical corrections:

HESSD
L 49 - groundwater-related and groundwater-dependent should be hyphenated

L 51 – missing a space after climate change

L 52 – the wording here is unclear – do previous studies not use numerical models but do use GCM data? Or do they use neither?

L 55 – the word focus is repeated twice in this sentence and again in the next – streamline if possible

- L 66 ease of use
- L 74 extraneous "and"
- L 76 the singular form of phenomena is phenomenon

L 89 – into

- L 90 strike "as a study site" it does not work grammatically and is not needed
- L 106 remove comma after distinct
- L 110 specify whether "large" refers to volume, rate, frequency, etc.?
- L 113 forest (unless you are referring to managed/planted forest with active timber harvesting)
- L 115 missing a period after Figure 1
- L 120 two commas
- L 219 flip order of extract and 5 km?
- L 229 Infoworks?

L 242 – spell out Nash-Sutcliffe Efficiency and Kling-Gupta Efficiency before abbreviating.

L 247 – 5 past and 19 future add up to 24 total not 25?
L 248 – introduce rainfall-runoff before abbreviating.

L 279 – define mOD before using abbreviation.

L 302 – grammar: either "which therefore leads us to conclude" or "which therefore indicates"

L 325 – missing space after Figure 5

L 494 - " property or human life"

---

## Author Comment (AC1) · 19 Nov 2020

Thank you for taking the time to review our paper and for you valuable comments. We have prepared a detailed response - it is better viewed through the supplemental pdf attached as the plain text below removes complicated equations and colours etc.

In the submitted manuscript, Morrissey et al use a semi-distributed karst model to estimate the possible impacts of climate change on a lowland karst system in Ireland both in terms of groundwater flooding and ecohydrology. Their model predicts that groundwater events that are currently considered extreme will increase in the future. In addition, a future shift of flood seasonality is simulated by the model, which the authors

suggest will likely affect ecological systems. Generally, the study is well-written and concise. Using a well-established karst model to estimate future groundwater flood frequency and ecohydrological implications is novel and of high interest for the readers of Hydrology and Earth System Sciences. As elaborated in the technical/specific remarks below, (1) comparing the RCMs' control periods simulations with meteorological observations may help to assess their trustworthiness. This comment is addressed fully below under the comment response "Methodology".

(2) estimating the uncertainty of the hydrological predictions may help avoiding wrong conclusions about moderate changes that are smaller than the prediction uncertainty This comment is addressed below under the comment "Karst Groundwater Model:" (3) A comparison with projected changes of karst processes of other karst model applications to assess future changes may permit some evaluation of the predicted changes of this study. This comment is addressed below under the comment "Results and Discussion" Overall, I am confident that the authors will be able addressing these point within the frame of minor revisions. Abstract: Some more information about the methods (model setup, climate scenarios) is necessary. The Abstract has been edited to include more detail about the methods, as well as an extra line to describe the issues of concern with flooding relating to infrastructure as well as ecology (point raised by Reviewer 2). Here is the revised text:

Lowland karst aquifers can generate unique wetland ecosystems which are caused by groundwater fluctuations that result in extensive groundwater-surface water interactions (i.e. flooding). However, the complex hydrogeological attributes of these systems linked to extremely fast aquifer recharge processes and flow through well-connected conduit networks often present difficulty in predicting how they will respond to changing climatological conditions. This study investigates the predicted impacts of climate change on a lowland karst catchment by using a semi-distributed pipe-network model of the karst aquifer populated with output from the high spatial resolution (4 km) COSMO-CLM regional climate model simulations for Ireland. An ensemble of

projections for the future Irish climate were generated by downscaling from five different global climate models (GCMs), each based on four Representative Concentration Pathways (RCP2.6, RCP4.5, RCP6.0 and RCP8.5) to account for the uncertainty in the estimation of future global emissions of greenhouse gases. The one dimensional hydraulic / hydrologic karst model incorporates urban drainage software to simulate open channel and pressurised flow within the conduits with flooding on the land surface represented by storage nodes with the same stage-volume properties of the physical turlough basins. The lowland karst limestone catchment is located on the west coast of Ireland and is characterised by a well-developed conduit dominated karst aquifer which discharges to the sea via intertidal and submarine springs. Annual above ground flooding associated with this complex karst system has led to the development of unique wetland ecosystems in the form of ephemeral lakes known as turloughs, however extreme flooding of these features causes widespread damage and disruption in the catchment. This analysis has shown that mean, 95th and 99th percentile flood levels are expected to increase by significant proportions for all future emission scenarios. The frequency of events currently considered to be extreme is predicted to increase, indicating that more significant groundwater flooding events seem likely to become far more common. The depth and duration of flooding is of extreme importance, both from an ecological perspective in terms of wetland species distribution and for extreme flooding in terms of the disruption to homes, transport links and agricultural land inundated by flood waters. The seasonality of annual flooding is also predicted to shift later in the flooding season which could have consequences in terms of ecology and land use in the catchment. The investigation of increasing mean sea levels, however showed that anticipated rises would have very little impact on groundwater flooding due to the marginal impact on ebb tide outflow volumes. Overall, this study highlights the relative vulnerability of lowland karst systems to future changing climate conditions mainly due to the extremely fast recharge which can occur in such systems. The study presents a novel and highly effective methodology for studying the impact of climate change in lowland karst systems by coupling karst hydrogeological models

with the output from high resolution climate simulations.

Introduction: Line 51: Typo "changeon" This typo has been corrected.

There is some work on groundwater level frequencies and climate change by Bloomfield & colleagues and some semi-distributed modeling of GW levels and climate change by Brenner et al (2018, NHESS), which may be useful for the state-of-the-art Thank you for this suggestion and we agree these studies are very relevant in describing the existing state-of-the-art. These references have now been added to the Introduction section – see amended extract from the Introduction below (amended reference list at end of this response).

Studies into the impacts of climate change have been carried out for the chalk aquifers of south-western England which have high porosity and are prone to karstification. Jackson et al. (2015) utilised a distributed ZOOMQ3D groundwater model of the Chalk aquifer with various emission scenario input data to investigate the predicted changes in groundwater levels. Brenner et al. (2018) conducted a further study of this chalk catchment and showed that projected climate changes may lead to generally lower groundwater levels and a reduction of exceedances of high groundwater level percentiles in the future. Chen et al. (2018) conducted a study into the effects of climate change on alpine karst using GCM data. However, the results of these studies are not directly relevant to lowland karst with significant groundwater-surface water interactions and associated eco-hydrological habitats (groundwater fed wetlands).

Regional Climate Modelling. This section is a short of a general review. Please either add study site specific information (which RCMs are available? how have they been established?) or move to Introduction. Comment is agreed - this section will now be moved to the Introduction in which it will fit better to help the flow of the paper.

Methodology Is it possible to compare the model ensembles of the historic/control time period with observed climate data? The authors will update the paper to include the following references in which this issue has been fully addressed: Nolan et al., (2017,

2020) and Flanagan et al. (2019, 2020) – see additional references list at the end of this document. The following text addresses this point and can either be incorporated into the main text or added as supplemental information: The RCMs were validated by downscaling ECMWF ERA-Interim reanalyses and the GCM datasets for multi-decadal time periods and comparing the output with observational data. Extensive validations were carried out to test the ability of the RCMs to accurately model the climate of Ireland. Figure XX(a) presents the annual observed precipitation averaged over the period 1981–2000. Figure XX(b) presents the downscaled ERA-Interim data as simulated by the COSMO5-CLM model with 4-km grid spacings. It is noted that the RCM accurately captures the magnitude and spatial characteristics of the historical precipitation climate, e.g. higher rainfall amounts in the west and over mountains. Figure XX(c) shows that the percentage errors range from approximately –30% to approximately +15% for COSMO5-CLM downscaled ERA-Interim data. The percentage error at each grid point (i, j) is given by: $per\_bias_{(i,j)}=100\times(bias_{(i,j)}/(OBS)_{(i,j)})$ (?.?) where $bias_{(i,j)}=(RCM)_{(i,j)}-(OBS)_{(i,j)}$ (?.?)

and the $(RCM)_{(i,j)}$ and $(OBS)_{(i,j)}$ terms represent the RCM and observed values, respectively, at grid point (i, j), averaged over the period 1981–2000. Figure XX(c) highlights a clear underestimation of precipitation over the mountainous regions. This is probably because the RCMs underestimate heavy precipitation; previous validations studies (e.g. Nolan et al., 2017) have demonstrated a decrease in RCM skill with increasing magnitude of heavy precipitation events. To assess the added value of high-resolution RCM data, and to quantify the improved skill of RCMs over the GCMs, precipitation data were compared with both RCM and GCM data for the period 1976–2005. Results, presented in Table YY, demonstrate improved skill of the RCMs over the GCMs. Moreover, an increase in grid resolution of the RCMs (from 18- to 4-km grid spacings) results in a general increase in skill. For an in-depth validation of the RCMs, please refer to Nolan et al. (2015, 2017, 2020), Flanagan et al. (2019, 2020) and Werner et al. (2019), the results of which confirm that the output of the RCMs exhibit reasonable and realistic features as documented in the historical data record and

consistently demonstrate improved skill over the GCMs. The results of these validation analyses confirm that the RCM configurations and domain size of the current study are capable of accurately simulating the climate of Ireland. (a) (b) (c) Figure XX. Mean annual precipitation for 1981–2000. (a) Observations, (b) COSMO5-CLM-ERA-Interim 4-km data and (c) COSMO5-CLM-ERA-Interim error (%).

30-year average annual rainfall MAE % error GCM GCM Data COSMO5-CLM-GCM 18 km COSMO5-CLM-GCM 4 km CNRM-CM5 16.5 14.1 11.8 EC-Earth (r12i1p1) 17.3 14.0 10.0 HadGEM2-ES 20.8 14.6 15.1 MIROC5 26.0 18.2 15.6 MPI-ESM-LR 25.1 24.8 21.6

Table YY. GCM and COSMO5-CLM Mean Absolute Error (%) uncertainty estimates through comparison with gridded observations for the period 1976–2005. For each metric, the best- and worst-performing scores are highlighted in green and red, respectively.

Lines 192-193: "GCM to 18 km to 4 km" Please rephrase into full sentence. This sentence has been rephrased.

The RCMs were driven by GCM boundary conditions with the following nesting strategy; GCM to 18 km and GCM to 4 km. For the current study, only 4 km grid spacing RCM data are considered. The higher resolution data allows sharper estimates of the regional variations of climate projections. The climate fields of the RCM simulations were archived at 3-h intervals.

Table 1: Not all RCPs are available for all of the models. How was this handled when working with the model ensembles? In each case, the future 30-year periods are compared with the past RCM period 1976-2005. the mean of three RCP2.6, five RCP4.5 and five RCP8.5 RCM projections were calculated. The RCP6.0 simulation comprises just one simulation so was compared directly with the past RCM period.

This methodology was summarised in the methodology section but the text above can

be added to the caption of Table 1 if required.

Figure 2: numbers in grey circles within figures not elaborated in caption Explanation of these numbers now added to the Figure caption (see response to Reviewer 2).

Karst Groundwater Model: Since the model parameters are estimated via calibration, it would make sense providing some estimate about their uncertainty especially since the model is used for prediction far in the future. Moderate predicted changes in flood frequency might be still within the envelop of the simulation uncertainty.

This is a valid issue and was one which we had considered when conducting this study. A thorough calibration and validation of the model was carried out which took over two years and required a large amount of field effort coupled with tedious model calibration. The process is described in detail in Morrissey et al. (2019). Whilst the model incorporates 15 flooding nodes which were calibrated using 2 years worth of field data – more long-term data were available at 5 locations where were the subject of numerous long term research projects. This allowed the model uncertainty to be compared over two separate overlapping time horizons: 2016-2018 and 2007-2018 at these locations. The resulting model uncertainty (using Nash-Sutcliff Efficiency values - NSE) was calculated for each location over these periods allowing an envelope of uncertainty to be determined – see Table 1 below. It can be seen that for the shorter and more recent calibration period the uncertainty for the model is approximately 5.2% (3% - 11%). For the longer validation period the uncertainty increases to 7.6% (3% - 14%). Using this envelope of uncertainty and comparing it to the projected changes in mean flood levels under climate change of 3.5% - 7.9% would suggest that the potential impacts of climate change are within the bounds of model uncertainty. Similarly, potential impacts of climate change on the average change in AEP flood level ranged between 2.92% - 10.07% and comparing this to the uncertainty envelope would appear to suggest a far more modest increase even in the High emissions scenario. Similar comparisons for the other key parameters such as flood duration would also yield results which appear to dampen the outcome of this study.

Table 1 Summary statistics for model uncertainty within the calibration & validation periods (Uncertainty = [1 − NSE] * 100) Location 2016 - 2018 Calibration Period 2007 - 2018 Validation Period Model uncertainty based on Volume (m3) Blackrock 3.00% 3.00% Caherglassaun 3.00% 2.00% Coole 8.00% 13.00% Coy 11.00% 14.00% Garryland 1.00% 6.00% Mean 5.20% 7.60%

However, the approach taken in this study circumvented the apparent issues outlined above by comparing the modelled past with the modelled future scenarios and by comparing within these model bounds for anomalies. The same model with uncertainty bounds as per Table 1 was used to both simulate the past RCM period (1976 − 2005) and the future time slice 2071 − 2100. By comparing the output from the RCM past and future simulations using the same calibrated model the error or bias within the model itself is accounted for and the anomalies between both periods represents the potential changes due to climate change. Other approaches for climate change modelling with GCM's use bias correction techniques to correct the simulated outputs for the past to correct the future and then utilise the differences between the two corrected datasets. This process can introduce further error given that bias correction for such models is an evolving field. The approach taken in this study has the advantage of eliminating the need for bias correction (which is a recognised method in the literature) and accounts for the karst model uncertainty.

We feel that this approach has in effect filtered out the karst model uncertainty thus the potential effects of climate change as reported in the paper do not need to account for model uncertainty. This approach was described in the paper between lines 198 − 206. The authors will add additional text similar to the above into the amended version of the paper to better explain the issue of the karst model uncertainty.

An overview of the simulations is presented in Table 1. Data from two time-slices, 1976–2005 (the control or past) and 2071–2100, were used for analysis of projected changes in the Irish climate by the end of the 21st-century. It must be noted that the full RCM simulations in fact covered the entire period 1976 − 2100 and these time slices

were simply used to make a past versus future comparison (Figure 2 shows results from the full simulation and not just the chosen time slices for this current study). The historical period was compared with the corresponding future period for all simulations within the same RCM-GCM group. This results in future anomalies for each model run; that is, the difference between future and past.

Results & Discussion: A general assessment on how much you trust the model projections is necessary. A comparison of the karst hydrological projections of this model with the hydrological projections of other karst models might be useful for that. Which processes are prediction to become more pronounced by this model and is this in agreement to the projected changes in hydrological behavior of other karst simulation models that were run with climate projections? In relation to uncertainty - see response above under Karst Groundwater Model. The following text can be added: Model uncertainty was compared to other karst models the reported uncertainty of our model (3 -14%) is comparable and within the same window when compared to other reported studies (e.g. Mudarra et. al., 2019, Sofia et. al, 2020)

The following text can be added to address comparison to other karst/climate studies: A study undertaken by Nerantzaki & Nikolaidis (2020) which was similar in nature (i.e use of GCM and RCM data with karst models) and indicated that a reduction of karst spring flow of between 14 - 25% could occur under climate change scenarios (authors used a blended rainfall spectrum from RCP2.6 and RCP8.5). This range is comparable to some of the results observed in this study. Similarly, other studies focused on the impacts of karst aquifer due to climate change utilise GCM/RCM and various emissions scenarios (Pardo-Igúzquiza et al., 2019 ) but are concerned with impacts to recharge (and spring water availability) and flooding/eco-hydrology are not considered. It is therefore difficult to provide direct comparisons with this current study, however the authors are confident the projections reported in this study are broadly in line with other international studies.

Do the projected karst hydrological changes agree with the conceptual understanding

of the lowland karst? The study catchment has been investigated by the authors for almost 20 years and we have gained valuable insight into how the system responds to various rainfall patterns and intensities. In this regard the outcomes of this study are in line with our conceptual understanding of the lowland karst. Slightly higher intensity rainfall events in the winter when the system is flooded will lead to higher flooding which is slower to drain to the sea. Text to reflect this stance can be added to the final version of the paper if the reviewer feels it is necessary.

The discussion about ecohydrology could be a bit more specific. Which negative specific consequences may occur? Which plants/animals are affected most? Maybe one or two specific examples would help. In parallel to this work there has also been a multi-disciplinary team of ecologists and water quality experts investigating these turloughs, with the results contained in (Irvine et al., 2018 and Waldren et al., 2015). This work has been continued in an ongoing project which has used a combination of more advanced spatial analysis in relation to long term hydrological data (over 298 years) as well as satellite data to define the ecohydrological metrics of relevance to different vegetation habitats. This is the subject of a new paper on ecohydrology which has just been submitted. Unfortunately, we can't provide a reference for this yet as it is still under review, but we will include some examples for specific vegetation communities that are sensitive to changes in the flooding regime. Also, we show some of the plots that have been generated from the recent work to back up the statistics presented (see below). The following text will be added to the Implications for mean and recurrent flood levels and eco-hydrology section from line 346.

The spatial distribution of different vegetation communities in such wetlands is intimately entwined with the hydrological conditions (flood duration, flood depth, time of year of flood recession etc.), which change on a gradient moving up from the base of the turloughs. These ecohydrological relationships have been researched in multidisciplinary studies on these turloughs investigating links between the fluctuating hydrological regime and vegetation habitats, invertebrates, soil properties, land use and water

quality (Kimberley et al., 2012; Irvine et al., 2018; Waldren et al., 2015) from which metrics have then be defined for the different key wetland habitats. For example, recent ecohydrological analysis the spatial distribution of vegetation habitats on four turloughs in this karst network (Blackrock, Coy, Garryland and Caherglassaun) over a 28 year period has revealed distinct differences between vegetation communities, from Eleocharis acicularis found at the base of the turlough typically experiencing 6 to 7 months of inundation per year compared to the limestone pavement community at the top fringes of the turloughs only flooded from 1 to 2 months per year. These differences in flood depth and duration are also reflected in a gradient of times across the early growing season (spring) when the communities emerge from the flood waters (and associated changes in air temperature and solar radiation). Other investigations on invertebrates in the turloughs (Porst and Irvine 2009, Porst et al., 2012) have shown that hydroperiod (flood duration) has a significant effect on macroinvertebrate taxon richness, with short hydroperiods supporting low faunal diversity. The study demonstrates how different colonisation cycles occur in response to the seasonal hydrological disturbances.

Figure XX. Annual flood duration spatial profiles for Blackrock turlough over 28-year period.

Figure XX. The statistics of flood duration as a metric across the range of turlough vegetation communities averaged over four turloughs over a 28-yr period.

Figure 4: please enlarge x-/y-axis label font sizes This is complete – see below.

Figure 4. – revised version to be incorporated into the paper.

Line 325: Typo "Figure 5illuminates" This typo has been corrected.

Please also note the supplement to this comment:
https://hess.copernicus.org/preprints/hess-2020-203/hess-2020-203-AC1-supplement.pdf

**Supplement:**

**Anonymous Referee #1** ()

In the submitted manuscript, Morrissey et al use a semi-distributed karst model to estimate the possible impacts of climate change on a lowland karst system in Ireland both in terms of groundwater flooding and ecohydrology. Their model predicts that groundwater events that are currently considered extreme will increase in the future. In addition, a future shift of flood seasonality is simulated by the model, which the authors suggest will likely affect ecological systems.

Generally, the study is well-written and concise. Using a well-established karst model to estimate future groundwater flood frequency and ecohydrological implications is novel and of high interest for the readers of Hydrology and Earth System Sciences. As elaborated in the technical/specific remarks below,

(1) comparing the RCMs' control periods simulations with meteorological observations may help to assess their trustworthiness.

This comment is addressed fully below under the comment response "Methodology".

(2) estimating the uncertainty of the hydrological predictions may help avoiding wrong conclusions about moderate changes that are smaller than the prediction uncertainty

This comment is addressed below under the comment "Karst Groundwater Model:"

(3) A comparison with projected changes of karst processes of other karst model applications to assess future changes may permit some evaluation of the predicted changes of this study.

This comment is addressed below under the comment "Results and Discussion"

Overall, I am confident that the authors will be able addressing these point within the frame of minor revisions.

**Abstract:**
Some more information about the methods (model setup, climate scenarios) is necessary.
The Abstract has been edited to include more detail about the methods, as well as an extra line to describe the issues of concern with flooding relating to infrastructure as well as ecology (point raised by Reviewer 2). Here is the revised text:

*Lowland karst aquifers can generate unique wetland ecosystems which are caused by groundwater fluctuations that result in extensive groundwater-surface water interactions (i.e. flooding). However, the complex hydrogeological attributes of these systems linked to extremely fast aquifer recharge processes and flow through well-connected conduit networks often present difficulty in predicting how they will respond to changing climatological conditions. This study investigates the predicted impacts of climate change on a lowland karst catchment by using a semi-distributed pipe-network model of the karst aquifer populated with output from the high spatial resolution (4 km) COSMO-CLM regional climate model simulations for Ireland. An ensemble of projections for the future Irish climate were generated by downscaling from five different global climate models (GCMs), each based on four Representative Concentration Pathways (RCP2.6, RCP4.5, RCP6.0 and RCP8.5) to account for the uncertainty in the estimation of future global emissions of greenhouse gases. The one dimensional hydraulic / hydrologic karst model incorporates urban drainage software to simulate open channel and pressurised flow within the conduits with flooding on the land surface represented by storage nodes with the same stage-volume properties of the physical turlough basins. The lowland karst limestone catchment is located on the west coast of Ireland and is characterised by a well-developed conduit dominated karst aquifer which discharges to the sea via intertidal and submarine springs. Annual above ground flooding associated with this complex karst system has led to the development of unique wetland ecosystems in the form of ephemeral lakes known as turloughs, however extreme flooding of*

*these features causes widespread damage and disruption in the catchment. This analysis has shown that mean, 95th and 99th percentile flood levels are expected to increase by significant proportions for all future emission scenarios. The frequency of events currently considered to be extreme is predicted to increase, indicating that more significant groundwater flooding events seem likely to become far more common. The depth and duration of flooding is of extreme importance, both from an ecological perspective in terms of wetland species distribution and for extreme flooding in terms of the disruption to homes, transport links and agricultural land inundated by flood waters. The seasonality of annual flooding is also predicted to shift later in the flooding season which could have consequences in terms of ecology and land use in the catchment. The investigation of increasing mean sea levels, however showed that anticipated rises would have very little impact on groundwater flooding due to the marginal impact on ebb tide outflow volumes. Overall, this study highlights the relative vulnerability of lowland karst systems to future changing climate conditions mainly due to the extremely fast recharge which can occur in such systems. The study presents a novel and highly effective methodology for studying the impact of climate change in lowland karst systems by coupling karst hydrogeological models with the output from high resolution climate simulations.*

**Introduction:**

Line 51: Typo "changeon"

This typo has been corrected.

There is some work on groundwater level frequencies and climate change by Bloomfield & colleagues and some semi-distributed modeling of GW levels and climate change by Brenner et al (2018, NHESS), which may be useful for the state-of-the-art

Thank you for this suggestion and we agree these studies are very relevant in describing the existing state-of-the-art. These references have now been added to the Introduction section – see amended extract from the Introduction below (amended reference list at end of this response).

*Studies into the impacts of climate change have been carried out for the chalk aquifers of south-western England which have high porosity and are prone to karstification. Jackson et al. (2015) utilised a distributed ZOOMQ3D groundwater model of the Chalk aquifer with various emission scenario input data to investigate the predicted changes in groundwater levels. Brenner et al. (2018) conducted a further study of this chalk catchment and showed that projected climate changes may lead to generally lower groundwater levels and a reduction of exceedances of high groundwater level percentiles in the future. Chen et al. (2018) conducted a study into the effects of climate change on alpine karst using GCM data. However, the results of these studies are not directly relevant to lowland karst with significant groundwater-surface water interactions and associated eco-hydrological habitats (groundwater fed wetlands).*

**Regional Climate Modelling.**

This section is a short of a general review. Please either add study site specific information (which RCMs are available? how have they been established?) or move to Introduction.

Comment is agreed - this section will now be moved to the Introduction in which it will fit better to help the flow of the paper.

**Methodology**

Is it possible to compare the model ensembles of the historic/control time period with observed climate data?

The authors will update the paper to include the following references in which this issue has been fully addressed: Nolan et al., (2017, 2020) and Flanagan et al. (2019, 2020) – see additional references list

at the end of this document. The following text addresses this point and can either be incorporated into the main text or added as supplemental information:

[revised manuscript text omitted]

This methodology was summarised in the methodology section but the text above can be added to the caption of Table 1 if required.

Figure 2: numbers in grey circles within figures not elaborated in caption
Explanation of these numbers now added to the Figure caption (see response to Reviewer 2).

**Karst Groundwater Model:**
Since the model parameters are estimated via calibration, it would make sense providing some estimate about their uncertainty especially since the model is used for prediction far in the future. Moderate predicted changes in flood frequency might be still within the envelop of the simulation uncertainty.

This is a valid issue and was one which we had considered when conducting this study. A thorough calibration and validation of the model was carried out which took over two years and required a large amount of field effort coupled with tedious model calibration. The process is described in detail in Morrissey et al. (2019). Whilst the model incorporates 15 flooding nodes which were calibrated using 2 years worth of field data – more long-term data were available at 5 locations where were the subject of numerous long term research projects. This allowed the model uncertainty to be compared over two separate overlapping time horizons: 2016-2018 and 2007-2018 at these locations. The resulting model uncertainty (using Nash-Sutcliff Efficiency values - NSE) was calculated for each location over these periods allowing an envelope of uncertainty to be determined – see Table 1 below.  It can be seen that for the shorter and more recent calibration period the uncertainty for the model is approximately 5.2% (3% - 11%). For the longer validation period the uncertainty increases to 7.6% (3% - 14%). Using this envelope of uncertainty and comparing it to the projected changes in mean flood levels under climate change of 3.5% - 7.9% would suggest that the potential impacts of climate change are within the bounds of model uncertainty.  Similarly, potential impacts of climate change on the average change in AEP flood level ranged between 2.92% - 10.07% and comparing this to the uncertainty envelope would appear to suggest a far more modest increase even in the High emissions scenario. Similar comparisons for the other key parameters such as flood duration would also yield results which appear to dampen the outcome of this study.

*Table 1 Summary statistics for model uncertainty within the calibration & validation periods (Uncertainty = [1 – NSE] * 100)*

| Location | 2016 - 2018 Calibration Period | 2007 - 2018 Validation Period |
|---|---|---|
| | Model uncertainty based on Volume ($m^3$) | |
| Blackrock | 3.00% | 3.00% |
| Caherglassaun | 3.00% | 2.00% |
| Coole | 8.00% | 13.00% |
| Coy | 11.00% | 14.00% |
| Garryland | 1.00% | 6.00% |
| **Mean** | **5.20%** | **7.60%** |

However, the approach taken in this study circumvented the apparent issues outlined above by comparing the modelled past with the modelled future scenarios and by comparing within these

model bounds for anomalies. The same model with uncertainty bounds as per Table 1 was used to both simulate the past RCM period (1976 – 2005)  and the future time slice 2071 – 2100. By comparing the output from the RCM past and future simulations using the same calibrated model the error or bias within the model itself is accounted for and the anomalies between both periods represents the potential changes due to climate change. Other approaches for climate change modelling with GCM's use bias correction techniques to correct the simulated outputs for the past to correct the future and then utilise the differences between the two corrected datasets. This process can introduce further error given that bias correction for such models is an evolving field. The approach taken in this study has the advantage of eliminating the need for bias correction (which is a recognised method in the literature) and accounts for the karst model uncertainty.

We feel that this approach has in effect filtered out the karst model uncertainty thus the potential effects of climate change as reported in the paper do not need to account for model uncertainty. This approach was described in the paper between lines 198 – 206. The authors will add additional text similar to the above into the amended version of the paper to better explain the issue of the karst model uncertainty.

*An overview of the simulations is presented in Table 1. Data from two time-slices, 1976–2005 (the control or past) and 2071–2100, were used for analysis of projected changes in the Irish climate by the end of the 21st-century. It must be noted that the full RCM simulations in fact covered the entire period 1976 – 2100 and these time slices were simply used to make a past versus future comparison (Figure 2 shows results from the full simulation and not just the chosen time slices for this current study).  The historical period was compared with the corresponding future period for all simulations within the same RCM-GCM group. This results in future anomalies for each model run; that is, the difference between future and past.*

**Results & Discussion:**
A general assessment on how much you trust the model projections is necessary. A comparison of the karst hydrological projections of this model with the hydrological projections of other karst models might be useful for that. Which processes are prediction to become more pronounced by this model and is this in agreement to the projected changes in hydrological behavior of other karst simulation models that were run with climate projections?
In relation to uncertainty - see response above under Karst Groundwater Model. The following text can be added:
*Model uncertainty was compared to other karst models the reported uncertainty of our model (3 -14%) is comparable and within the same window when compared to other reported  studies (e.g. Mudarra et. al., 2019, Sofia et. al, 2020)*

The following text can be added to address comparison to other karst/climate studies:
*A study undertaken by Nerantzaki & Nikolaidis (2020) which was similar in nature (i.e use of GCM and RCM data with karst models) and indicated that a reduction of karst spring flow of between 14 - 25% could occur under climate change scenarios (authors used a blended rainfall spectrum from RCP2.6 and RCP8.5). This range is comparable to some of the results observed in this study. Similarly, other studies focused on the impacts of karst aquifer due to climate change utilise GCM/RCM and various emissions scenarios (Pardo-Igúzquiza et al., 2019 ) but are concerned with impacts to recharge (and spring water availability) and flooding/eco-hydrology are not considered. It is therefore difficult to provide direct comparisons with this current study, however the authors are confident the projections reported in this study are broadly in line with other international studies.*

Do the projected karst hydrological changes agree with the conceptual understanding of the lowland karst?

The study catchment has been investigated by the authors for almost 20 years and we have gained valuable insight into how the system responds to various rainfall patterns and intensities. In this regard the outcomes of this study are in line with our conceptual understanding of the lowland karst. Slightly higher intensity rainfall events in the winter when the system is flooded will lead to higher flooding which is slower to drain to the sea. Text to reflect this stance can be added to the final version of the paper if the reviewer feels it is necessary.

The discussion about ecohydrology could be a bit more specific. Which negative specific consequences may occur? Which plants/animals are affected most? Maybe one or two specific examples would help. In parallel to this work there has also been a multi-disciplinary team of ecologists and water quality experts investigating these turloughs, with the results contained in (Irvine et al., 2018 and Waldren et al., 2015). This work has been continued in an ongoing project which has used a combination of more advanced spatial analysis in relation to long term hydrological data (over 298 years) as well as satellite data to define the ecohydrological metrics of relevance to different vegetation habitats. This is the subject of a new paper on ecohydrology which has just been submitted. Unfortunately, we can't provide a reference for this yet as it is still under review, but we will include some examples for specific vegetation communities that are sensitive to changes in the flooding regime. Also, we show some of the plots that have been generated from the recent work to back up the statistics presented (see below). The following text will be added to the **Implications for mean and recurrent flood levels and eco-hydrology section** from line 346.

*The spatial distribution of different vegetation communities in such wetlands is intimately entwined with the hydrological conditions (flood duration, flood depth, time of year of flood recession etc.), which change on a gradient moving up from the base of the turloughs. These ecohydrological relationships have been researched in multidisciplinary studies on these turloughs investigating links between the fluctuating hydrological regime and vegetation habitats, invertebrates, soil properties, land use and water quality (Kimberley et al., 2012; Irvine et al., 2018; Waldren et al., 2015) from which metrics have then be defined for the different key wetland habitats. For example, recent ecohydrological analysis the spatial distribution of vegetation habitats on four turloughs in this karst network (Blackrock, Coy, Garryland and Caherglassaun) over a 28 year period has revealed distinct differences between vegetation communities, from Eleocharis acicularis found at the base of the turlough typically experiencing 6 to 7 months of inundation per year compared to the limestone pavement community at the top fringes of the turloughs only flooded from 1 to 2 months per year. These differences in flood depth and duration are also reflected in a gradient of times across the early growing season (spring) when the communities emerge from the flood waters (and associated changes in air temperature and solar radiation). Other investigations on invertebrates in the turloughs (Porst and Irvine 2009, Porst et al., 2012) have shown that hydroperiod (flood duration) has a significant effect on macroinvertebrate taxon richness, with short hydroperiods supporting low faunal diversity. The study demonstrates how different colonisation cycles occur in response to the seasonal hydrological disturbances.*

[Figure]

**Figure XX.** Annual flood duration spatial profiles for Blackrock turlough over 28-year period.

[Figure]

**Figure XX.** The statistics of flood duration as a metric across the range of turlough vegetation communities averaged over four turloughs over a 28-yr period.

Figure 4: please enlarge x-/y-axis label font sizes
This is complete – see below.

[Figure]

Figure 4. – revised version to be incorporated into the paper.

Line 325: Typo "Figure 5illuminates"
This typo has been corrected.

**Anonymous Referee #2**

Questions to address:
1. Does the paper address relevant scientific questions within the scope of HESS?
**Yes.** What are the impacts of climate change on flooding in lowland karst areas?
No response.

2. Does the paper present novel concepts, ideas, tools, or data? **Yes.** Existing modeling tools and datasets are integrated and analyzed in a novel way.
No response.

3. Are substantial conclusions reached? **Yes.** Flooding is likely to increase significantly (both in magnitude and duration) in the studied catchments due to climate-induced changes in precipitation patterns. Sea level rise is not likely a factor in increasing flood risk in the study catchments.
No response.

4. Are the scientific methods and assumptions valid and clearly outlined? **Yes.** An ensemble of regional climate models was used to generate inputs to a pipe-flow model of groundwater flow in the study catchment. Thorough statistical analyses demonstrate that the model results are significant. No response.

5. Are the results sufficient to support the interpretations and conclusions? **Yes.**
No response.

6. Is the description of experiments and calculations sufficiently complete and precise to allow their reproduction by fellow scientists (traceability of results)? **Yes.**
No response.

7. Do the authors give proper credit to related work and clearly indicate their own new/original contribution? **Yes.**
No response.

8. Does the title clearly reflect the contents of the paper? **Somewhat.** This study does an excellent job of thoroughly analyzing the potential changes in flooding patterns in lowland karst areas due to climate change, but there is no in-depth analysis of the impacts on the ecohydrology. The one paragraph discussing ecohydrology at the end of the paper includes no citations, data, or analysis. I would therefore recommend changing the title to "Impacts of climate change on groundwater flooding in lowland karst".
This has been addressed in responses below.

9. Does the abstract provide a concise and complete summary? **No.** There is a misleading emphasis on ecology. The paper focuses on flooding, therefore, the abstract should as well. A sentence should be added indicating why flooding is a concern.
The first sentence is also misleading – this paper is focused on turloughs, and the abstract should therefore reflect that by beginning with a clear one-sentence description of turloughs. However, the rest of the abstract is quite good.
The abstract has been edited to include a sentence linking ecology to hydrology in these intermittent wetlands (as well as the additional detail on the methodology as requested by Reviewer 1),.

10. Is the overall presentation well structured and clear? **Mostly.** The motivation for studying flooding should be presented at the beginning – the authors indicate that flooding is a concern but do not explain why until the second-to-last section. I would recommend moving the bulk of the description of why flooding is harmful to the introduction, possibly under the sub-heading "Motivation". Also, the introduction emphasizes drought and ecological impacts, but drought is not discussed at all in the rest of the paper, and ecological impacts are discussed only briefly.
These comments have now been addressed (see below) and also a better discussion on the aspect of droughts / dry periods for these intermittent wetlands is included in the paper.

11. Is the language fluent and precise? **Somewhat.** The manuscript would benefit
from a thorough reading for grammar, typos, and consistency. There are unnecessarily
awkward third-person sentence constructions.
Significant reviewing and edits have been undertaken – see responses above and below.

12. Are mathematical formulae, symbols, abbreviations, and units correctly defined
and used? **Mostly.** Several abbreviations and units are used without being first defined.
Abbreviations are not always consistent.
This has now been addressed – see below.

13. Should any parts of the paper (text, formulae, figures, tables) be clarified, reduced,
combined, or eliminated? The section on ecohydrology is lacking references and specificity,
and should therefore either be expanded into a full, well-referenced discussion or
eliminated.
The ecohydrology section has been redrafted  - see below.

14. Are the number and quality of references appropriate? **Yes** (except for abovementioned
exception in the ecohydrology section).
The ecohydrology section has been redrafted  - see below.

15. Is the amount and quality of supplementary material appropriate? **Not applicable**.
No response.

**General comments:**
The authors present a well-thought-out modeling study with methods that may be applicable
to other lowland karst catchments vulnerable to climate change impacts, and with findings that are
likely to be of great interest to planners in responding to climate related stresses such as flooding. The
article is generally well-organized and clear, but requires a thorough grammar/typo/consistency
revision. My only substantive critique is that the ecohydrology section is quite thin and should either
be overhauled or eliminated. Finally, the paper would be more broadly relevant and interesting if it
also incorporated a spatial analysis of flooding (which may or may not be possible given the modeling
setup). However, if it is possible, I believe it would be well worth the authors' time to expand the scope
of the analysis slightly to include this (see more detailed comments below).

All of these points have been addressed in the more detailed comments below.

**Specific comments: individual scientific questions/issues**

**1. Introduction:**
a. Line 45: Please describe the projected shifts in precipitation patterns – Increase/
decrease? Change in seasonality? Change in spatial distribution? Intensity?
Frequency?
The mid-century precipitation climate of Ireland is expected to become more variable with substantial
projected increases in both dry periods and heavy precipitation events (Nolan 2017, 2020). These
studies show that substantial decreases in precipitation are projected for the summer months, with
reductions ranging from 0% to 11% for the RCP4.5 scenario and from 2% to 17% for the RCP8.5
scenario. Other seasons, and over the full year, show relatively small projected changes in
precipitation. The frequencies of heavy precipitation events show notable increases over the year as
a whole and in the winter and autumn months, with projected increases of 5–19%. The number of
extended dry periods is also projected to increase substantially by the middle of the century over the

full year and for all seasons except spring. The projected increases in dry periods are largest for summer, with values of +11% and +48% for the RCP4.5 and RCP8.5 scenarios, respectively.

b. Line 58 & 68: The terms habitat and ecosystem are used interchangeably throughout the text. Usually habitat describes conditions appropriate for a specific organism or type of organism of interest, while ecosystem describes the entire biotic and abiotic community. Unless the authors have a specific organism/species in mind that is at risk or of particular local significance, it is best to use the term "ecosystem" rather than "habitat". Also, it is not clear here what an "eco-hydrological habitat" means. Is this referring to a groundwater-dependent ecosystem? If so, it is best to define and use the latter term consistently (it is introduced as an abbreviation on line 68 but then not
used again). If not, please define "eco-hydrological habitat" and specify the organism for which this habitat is present. Or just consistently use the term "groundwater-fed wetland" if that is the particular ecosystem present in the study area.

We agree that the term habitat should be used to describe conditions appropriate for a specific organism or community of interest, while ecosystem describes the entire biotic and abiotic community and have gone through the document to clarify this where our use may have come across as slightly ambiguous by replacing habitat by ecosystem (for example in lines 18, 38, 61, 95, 334 etc.). We do want to retain the word habitat in some places as we do have specific vegetation communities in mind that have been studied (as detailed in the earlier response to reviewer 1).

By ecohydrological habitat we are referring to the envelope of hydrological conditions (flood duration, flood depth, time of year of flood recession etc.) which support different vegetation communities in such groundwater-dependent ecosystems, which change on a gradient moving up from the base of the turloughs. This will be clarified in the Introduction. We have been doing a lot of work on these types of relationships over the past few years on these turloughs which is the subject of another paper just submitted. However, we have now included a lot more information on these ecohydrological relationships, as is detailed in the earlier response to reviewer 1.

c. Line 68: Please include a sentence explaining why droughts and floods threaten groundwater-dependent ecosystems.

An additional line has been added at this point as follows, but then much more information about the interlinks between hydrology and ecology is presented later in the paper, as per the earlier response to Reviewer 1.

*The spatial distribution of different vegetation communities in such wetlands is linked to the prevalent hydrological conditions (flood duration, flood depth, time of year of flood recession etc.) (Irvine et al., 2018), and therefore the impacts of predicted climate change on the hydrological dynamics will have a direct impact on the different vegetation habitats.*

d. Line 71: Please include a sentence or clause explaining why karst models are more difficult to couple with GCMs/RCMs.

Quote from Hartman (2017) paper in which he justifies this as…more heterogenous and non-stationary systems

e. Line 76: Please briefly define groundwater flooding vs. fluvial flooding.

The authors will add the following text at line 76:

*Whilst fluvial models (models which simulate flow with rivers) are relatively straightforward to calibrate and couple with the output from Global or Regional Climate Models, groundwater (and specifically karst) models can be more difficult to employ in such a manner, particularly in terms of assessing the resultant output (Hartmann, 2017).*

In addition, at the beginning of the following paragraph (line82) the following sentence will be added:

*Groundwater flooding occurs when the water table rises above the land surface flooding areas often for prolonged periods (often many weeks or months). This compares to fluvial flooding which occurs when river (or lake) systems overflow their banks and flow into the surrounding lands. Fluvial flooding typically occurs in a sudden (or dramatic) and sometimes dangerous manner following intense rainfall and dissipates relatively quickly (days).*

f. Line 82: Please include at least a sentence explaining what types of damage and disruption are caused by flooding – infrastructure damage, cutting off transportation access, destruction of homes, preventing planting or harvesting of crops? Please expand on how is it different from fluvial flooding. Please also briefly explain the potential impacts of drought on human society (infrastructure, agriculture, water supply, etc.).

The flooding which occurred in the catchment in 2009 was the most severe on record, until it was surpassed in many areas by the events of 2015. The two most recent flood events led to considerable damage, disruption and hardship for local residents and farmers. Over 24 km$^2$ of land was flooded for up to 6 months with many residents and farms cut-off due to roads being impassable for extended periods. The development of the karst model (used in this assessment) allowed a thorough assessment of flood risk throughout the Gort Lowlands study area to be concluded for a 1 in 100 year (1% AEP flood event). The following flood risk has been identified for this flood event:

- 50 No. residential properties flood and a further 23 No. are at high flood risk for prolonged durations.
- >463 ha of agricultural lands flood for periods greater than 3 months.
- 175 No. residential properties and 46 No. non-residential properties, including dairy farms, are cut-off due to prolonged flooding of all road access points for periods > 3months.
- The national M18 motorway connecting two regional cities and many other secondary/ regional and local roads flood for prolonged durations (>2weeks).
- The regional Limerick-Athenry Railway line will flood for a period of over 25 days.

g. Line 83: This section is a bit scattered. It would benefit from being restructured. Either discuss the effects of both flooding and drought on groundwater-dependent ecosystems, and then discuss the effects of both flooding and drought on human society, or have a paragraph on drought and a paragraph on flooding, with human and ecosystem impacts of each. I would suggest focusing on flooding and human impacts, since the ecosystem impacts discussion later in the paper is not as well fleshed out and there is no substantial discussion later of drought.

This section has been restructured as follows:

*The phenomena of groundwater flooding in general has become more reported as a natural hazard in recent decades following extensive damage to property and infrastructure across Europe in the winter of 2000-2001 (Finch et al., 2004, Pinault et al., 2005, Hughes et al., 2011). Significant groundwater flooding also occurred in the UK at Oxford (2007) and at Berkshire Downs and Chilterns (2014) and in Galway, Ireland in 2009 & 2015/2016 (Naughton et al., 2017). Whilst it has been reported that groundwater flooding rarely poses a risk to human life, this form of flooding is known to cause damage and disruption over a long duration, particularly when compared to fluvial flooding (Morris et al., 2008, Cobby et al., 2009). Climate change is also likely to further exacerbate extreme droughts (Murphy et al., 2019) and their frequency and persistence must be quantified if resource planning and protection are to be implemented. Equally, as discussed, the effects of changes in hydrological regimes to wetland ecosystems can be significant; for example, recent studies (Spraggs et al., 2015, Noone et al., 2017) have attempted to quantify the frequency and extent of historic droughts to better understand their recurrence interval and thus assess the resilience of different impacted wetland ecosystems.*

**2. Study Catchment**

a. Figure 1:

i. Please either include topography underlay or indicate flow directions – it is not clear what the predominant flow patterns are from the map as is. ii. Please include an inset showing the location of the study catchment within the country, ideally with major karst areas in the country indicated (maybe use the World Karst Map freely available GIS data?).

iii. Please label all locations referenced in the text (Kinvara Bay, Galway Bay, Gort Lowlands, Galway Lowlands, Slieve Aughty Mountains, etc.).

iv. Does the study area have a name? If so, include it.

v. The caption says that model nodes are represented, but they are not clearly indicated or visible on the map?

A revised version of Figure 1 has been produced (see below) which addresses all of the point above.

[Figure]

Figure 1. – revised version to be incorporated into the paper.

b. Line 100 & 120: Though I am not very familiar with turloughs, my understanding is that they are a type of polje. This should be clearly stated when turloughs are defined and introduced, so that readers familiar with karst generally but not Irish karst specifically will be able to place these features in the context of other karst systems.

The definition of turloughs is currently split between lines 100 and 120 and should be condensed into one section. Also, the current wording at line 120 makes it seem as though the term turlough refers only to the lake (when present), while line 100 makes it seem as though it refers to the depression even when dry. Please clarify.

Yes, turloughs are very similar to poljes in the way that they operate hydraulically – just slightly different from a geomorphological perspective. The complete definition of turloughs is now included from line 100 as follows (with a reference to poljes too) with the excess text in line 120 removed.

*Turloughs occur in glacially formed depressions in karst, which intermittently flood on an annual cycle via groundwater sources and have substrate and/or ecological communities characteristic of wetlands. Geomorphologically they are a variant on a polje which are generally larger and more flat-bottomed enclosed depressions in karst landscapes (Ford and Williams, 2007).*

c. Line 122-126: Please specify what type of damage was caused by flooding (see earlier comment).
See earlier response.

**3. Regional Climate Modelling**
a. This is a very nice description of model downscaling!
No response (thanks!)

**4. Climate Models and Methods**
a. It would be helpful to have a table or bullet points giving a brief summary of the features, strengths, and weaknesses of the five global datasets being used. Possibly this could be included in Table 1.
A validation analysis of the relative skill of the GCMs (and corresponding RCM downscaled data) in the simulation of precipitation will now be included either in the main paper or as supplemental information – see response to Reviewer 1 comment above under "Methodology".

b. Please give a brief explanation of the RCPs – what does the number attached mean ($CO_2$ concentration?), and what does it represent (low, medium, high emissions scenario?).
Throughout the rest of the paper, please use consistent terminology and color schemes for the RCPs. The text's readability would be improved if, once the different RCPs were introduced, they were then consistently referred to as low, medium, and high emission scenarios, and labelled as such in the figures and tables. As it is, they are alternately referred to by a confusing range of abbreviations and descriptive phrases, and are represented by different colors in each figure. The clearest visual representation would be a sequential color gradient from low to high emissions. Again, possibly the definitions could be included in Table 1.
The Representative Concentration Pathways (RCPs) are greenhouse gas concentration trajectories adopted by the IPCC. The RCPs are focused on radiative forcing – the change in the balance between incoming and outgoing radiation via the atmosphere caused primarily by changes in atmospheric composition – rather than being linked to any specific combination of socioeconomic and technological development scenarios. There are four such scenarios (RCP2.6, RCP4.5, RCP6.0 and RCP8.5), named with reference to a range of radiative forcing values for the year 2100 or after, i.e. 2.6, 4.5, 6.0 and 8.5W/m2, respectively (Moss et al., 2010; van Vuuren et al., 2011).

The above text can be added to the main paper above Table 1 if required or to the supplemental information.

c. Figure 2: This is a nice visualization of the climate models.
i. Are these the means of all 5 GCMs for each RCP? Please clarify which scenarios are being displayed.
Now clarified – see revised Figure 2 below.

**RCM Ensemble Winter Rainfall Projections**

[Figure]

***Figure 2 (revised) RCM Ensemble Projections of Mean Winter Rainfall (%). The individual ensemble percentage projections are calculated as 100×(future-past)/past. In each case, the future 30-year periods are compared with the past RCM period 1976-2005. The figure presents the mean of three RCP2.6 (Low), five RCP4.5 (Med), one RCP6.0 (Med/High) and five RCP8.5 RCM (High) projections. The numbers included on each plot are the minimum and maximum projected changes, displayed at their locations. (refer to Figure 1 for location of study catchment)***

ii. Please label the colorbar more specifically – percent increase or decrease from observed mean historical precipitation? Or is it from the mean modeled precipitation across the modeled past period (1975-2005)? Not clear.

Now clarified – see revised Figure 2 above.

iii. Correct the RCP labels to include the decimal point. Adding high/medium/low labels like in Fig. 3 would be helpful. See previous comment about consistent naming.

Now clarified – see revised Figure 2 above.

iv. Please define/explain the small numbers in gray bubbles in the caption.

Complete – see revised caption above.

v. Please include the outline or point location of the study catchment within the larger maps of Ireland.
Caption now refers reader back to Figure 1.

**5. Karst Groundwater Model**
a. Line 228: Please explicitly state who developed the model. The current third-person passive construction muddies authorship.
Following rephrase proposed:
A semi-distributed pipe network model of the Gort lowlands has been developed **by the authors** using urban drainage software (Inforworks ICM by Innovyze).

**6. Results & Discussion**
a. Line 266: Please specify the direction/type of bias (overpredict/underpredict? Etc.)
See response to Reviewer 1 under "Methodology".

**7. Statistical analysis**
a. Line 274: What are typical ranges of p values and what values would indicate statistical insignificance?
p-values less than 0.05 are consider statistically significant. A p-value of ~0 is highly statistically significant. p-values >0.05 are considered statistically insignificant.

b. Figure 3: Please label axes more clearly. Y-axis: Probability of non-exceedance F(x). X-axis: Spell out whatever mOD is an abbreviation for. In caption, state primary takeaway from figure: Probability of non-exceedance is lower for future climate scenarios compared to past, therefore flooding is more likely in all future scenarios.
Y-axis now relabelled more clearly as per the comment. Consistent colouring and labelling have now been revised for all plots from the model output throughout the paper.

[Figure]

Figure 3 (revised): Comparison of the non-parametric Cumulative Distribution Function (CDF) plots for the past and future RCM scenarios using the MPI-ESM-LR GCM datasets at Coole Turlough [the y-axis shows the probability F(x) of a particular flood stage (mOD) being less than or equal to x]

Why was Coole Turlough chosen? Is it representative of other turloughs in the study area?
Coole turlough is one of the key central turloughs in the region (and yes, representative of others)

c. L 292: Please include a brief discussion of possible reasons why the HADGEM2-ES and MIROC5 datasets might predict little to negative change in flood levels.
The reason for these results is linked to the factors which impact karst flooding (e.g., which season, dry/wet event impacts, winter vs summer, evapotranspiration vs precipitation, etc). The karst system responds to previous cumulative rainfall along with existing flood level so the pattern of rainfall is crucial to the level and extent of flooding. Given that the GCM/RCM data are randomised, the response of the karst model to the varying inputs will range. The use of ensembles mitigates this potential area of uncertainty and gives a better indication of likely future scenarios and in this regard we feel that our approach is robust.

d. Figure 4: Why was Cahermore Turlough chosen? Is it representative of other turloughs in the study area? Why do Fig. 3 & 4 represent different turloughs? See previous comments about labeling and coloring of RCPs.
Different turloughs were shown for different sections to highlight the impacts at different locations across the catchment. Only turloughs representative of the catchment were chosen (Coole/Caherglassaun).

See previous responses and revision below – consistent colouring and labelling has now been revised.

[Figure]

Figure 4. – revised version to be incorporated into the paper.

**8. Implications for mean & recurrent flood levels & eco-hydrology**
a. Table 2: For planning purposes, it would be useful to know what the mean flood stage is for each location and scenario as well.

A table with mean and max depths can be added however the authors are not convinced at how relevant this is to the overall study which relates to climate change impacts. The projected changes were the focus of this study and not the existing system operation.

b. Line 335: Please discuss the effects of late-season flooding in more detail. Roughly how much farmland is in the turlough-adjacent flood zone? Are there any studies of flood impacts on wet

grasslands and the general ecology of these systems? Also, please define "knock-on effect" or use a more widely understood phrase.

In this lowland karst area, the majority of adjacent land to the turlough is farmland. As per the earlier response to Reviewer 1, yes there have been a lot of studies on the link between hydrological flooding regimes and the ecology of the systems (particularly different vegetation communities) which has now been included in an additional paragraph just after this text in line 335, which will put the statement in much more context. The phrase knock-on has been replaced with "associated impact"

c. Figure 5: This is a nice figure. It would read more clearly if the colors and labels for the RCPs matched other figures (see previous comment). Please explain why Coole Turlough was chosen – is it representative of the others?

See previous responses and revision below – consistent colouring and labelling has now been revised.

[Figure]

Figure 5: – revised version to be incorporated into the paper.

d. Line 346: This sentence or something like it should be included in the abstract and in the introduction, to explain why flooding is a concern. Even better would be to provide estimates of how much agricultural land, how many residences, and how much major infrastructure is in the affected area.

The abstract has now undergone a large edit (in response to Reviewer 1's comments) and this point has now been included as well. Estimates of the amount of infrastructure affected has been addressed in a previous response.

e. Line 376: Please provide some sort of evidence for this claim.

These changes in flood durations and the recurrence of flooding above established "norms" will undoubtedly have significant impacts for turlough eco-hydrology.

This should now be addressed by the earlier additional information on the development of ecohydrological metrics for different vegetation communities that has been going on in parallel to this work. The sentence has been rewritten as follows to make this more accurate.

*These changes in flood durations and the recurrence of flooding outside of the determined ecohydrological metric envelopes will undoubtedly have significant impacts for turlough eco-hydrology.*

f. Figure 6: It is hard to tell apart RCP 4.5 and 8.5 because the colors are so similar. See comments about consistent color use across figures.

See previous responses and revision below – consistent colouring and labelling has now been revised.

[Figure]

Figure 6: – revised version to be incorporated into the paper.

**9. Implications for extreme flood events**
a. Figure 7: See comments for Figure 3.

See previous responses and revision below – consistent colouring and labelling has now been revised.

[Figure]

Figure 7: – revised version to be incorporated into the paper.

b. Line 415: Be cautious of stating that something is definitively proven, especially statistically. The K-S test indicates that the results are statistically significant. These are two different things.
Language now modified – see below:

*Given this test **indicates** that a future trend exists, the 95th and 99th percentile values at each model node were then calculated for each of the ensemble RCM simulations and the ensemble average percentage change between each of the past and future sceneries was used to determine the ensemble average across the entire catchment (see Table 4).*

**10. Impact of rising mean tide levels**
a. This is interesting! Would you expect areas with small to no tidal fluctuations to see
more of an impact from rising sea levels?
The model did not predict any significant impact from rising sea levels.

*This indicates that the differences between the distributions with mean sea level increases are statistically insignificant and that rises in mean sea levels of up to 1.05 m will have little impact in this karst catchment over and above the impacts of changing climate.*

**11. Groundwater flooding**
a. For planning purposes, it would be particularly interesting to see predictions of the spatial distribution of flooding. Is it possible to include some analysis and maps of the spatial extent of peak

flooding under different scenarios? What about maps of the catchment showing number of flooded days per year at each location? Or the last day of spring flooding? If the modeling approach presented in this paper could generate such maps for this and other catchments, it would be a powerful adaptation planning tool, and I think it would be well worth the time to add these analyses.

This approach is under way as part of a national groundwater flooding project and a team of researchers are implementing our approach nationally. We have completed such an exercise for the 1% AEP flood for our study catchment to compare with the same 1% AEP flood using the RCP4.5 (Med) ensemble results from this study and have completed the associated spatial exercise. See additional proposed Figure to be included in the revised paper. Accompanying text for this Figure will be:

*The spatial extent of the 1% AEP flood for the study catchment was carried out and compared to a similar map produced for the same flood using the RCP4.5 (Med) ensemble results – see Figure XX. The 1% AEP flood predicts that 24.18km² will be flooded during the peak. This compares to 29.77km² inundated during the RCP4.5 (Med) scenario (a 23% increase). |It must be noted that Figure XX only includes the food extents of the subject model and flooding from other sources (not simulated) would also likely occur during such an event.*

[Figure]

Figure XX – comparison of the spatial extent of the 1% AEP flood event for the study catchment and the associated increases predicted during the RCP4.5 (Med) ensemble scenario.

**12. Eco-hydrology**

a. This section does not cite any references to support the claims made. There are several interesting and valuable ideas, but they are not discussed in much depth, nor are they supported by evidence. I would therefore recommend either removing this section and the discussion of flooding impacts on the turlough ecosystems entirely, or taking the time to develop it properly (the latter option would be an excellent contribution and I hope that the authors will choose to explore this in more depth).

As per the response to earlier comments we would like to keep the discussion to ecohydrology int his paper as ther has bene a lot of research going on in parallel on ecohydrological metrics of different vegetation communities on these turloughs There has been a lot of additional information added now the earlier discussion on ecohydrology, as per the previous responses, which has included several additional references to new work in this area. Hence, it is felt that at this stage of the conclusion section of the paper, a lot fo the comments made should now be clearer and in context. However, edits have been made to this section as follows to address the point raised.

*Ecosystems which rely on groundwater to sustain wetland conditions are at particular risk to changes in inundation fluctuation regimes brought about by climate change. This study has shown that the pattern of flooding at turloughs in the west of Ireland is likely to change significantly with higher mean flood levels over longer durations. Different unique habitats have developed under such cyclical envelopes of hydrological conditions, presenting a spatial gradient of different communities that can exist under the different conditions moving up from the base of the turlough. Hence, the results of this climate change study predict that a change in the hydrological regime is likely to cause associated changes in the location and extent of these habitat zones within turloughs. Furthermore, some of these habitats may be at threat due to the predicted shift in the seasonality of flooding to later in the hydrological year, causing a delay in the critical early growing season for wetland grasses and flora. Ongoing studies have been investigating the differences in prevailing air temperature and solar radiation for the vegetation communities across the turloughs as they come out of the winter flood regime at different times and are first exposed to air in the spring. The increase in more extreme events could also have a detrimental impact to fringing habitats which develop along the perimeter of these sites (typically woody shrubs and trees or limestone pavement communities) which would be severely impacted were they to become flooded on a more regular basis. An argument could be made that the habitat zones could simply be shifted upwards in elevation, essentially expanding the extents of the wetlands. However, given that turloughs are often located within defined basins, the room for their "growth" is constrained and the loss of some habitat is likely to be unavoidable. For other similar groundwater dependent ecosystems in similar climate zones in karst such as fens the implications of fluctuations in future groundwater levels and flows are equally significant.*

b. See previous comments about terminology with respect to "habitat", "ecosystem", "groundwater-dependent ecosystem", etc.
These terms are now used more accurately as per previous comment.

c. There is no discussion of drought in the conclusions. Drought should therefore either not be mentioned in the introduction, or it should be made clear that drought is not within the scope of this paper.
The drought mentioned in the introduction was in relation to other studies on wetlands (as well as water resources) in general, not specifically to turloughs. The concept of drought for a turlough could either mean longer dry periods (growing seasons) due to droughts in the winter (which is not what the predictions are telling us) or drier conditions in the summer period when they are empty of water but maybe not enough rainfall to support the growth of vegetation from a soil moisture perspective. We agree that the second point has not been addressed in this study (or really evaluated in studies as far as we know for turloughs) and so any reference to this in relation to our study has been removed as suggested.

d. What is the potential transferability of this approach to other locations? This would be worth discussing briefly somewhere in the conclusion.
The final line of the paper has been edited to include this point as follows.
*In the wider context, this study has shown that the use of complex transient groundwater models with the output from RCM models can provide specific and targeted information on the likely effects of*

*climate change on groundwater levels, flooding and eco-hydrology. More specifically this methodology can clearly be transferred to study other karst based GWDTEs such as calcareous fens and poljes.*

All of the following typos have now been corrected.
**Technical corrections:** (all corrected).
L 49 – groundwater-related and groundwater-dependent should be hyphenated
L 51 – missing a space after climate change
L 52 – the wording here is unclear – do previous studies not use numerical models but do use GCM data? Or do they use neither?
L 55 – the word focus is repeated twice in this sentence and again in the next – streamline if possible
L 66 – ease of use
L 74 – extraneous "and"
L 76 – the singular form of phenomena is phenomenon
L 89 – into
L 90 – strike "as a study site" – it does not work grammatically and is not needed
L 106 – remove comma after distinct
L 110 – specify whether "large" refers to volume, rate, frequency, etc.?
L 113 – forest (unless you are referring to managed/planted forest with active timber harvesting)
L 115 – missing a period after Figure 1
L 120 – two commas
L 219 – flip order of extract and 5 km?
L 229 – Infoworks?
L 242 – spell out Nash-Sutcliffe Efficiency and Kling-Gupta Efficiency before abbreviating.
L 247 – 5 past and 19 future add up to 24 total not 25?
L 248 – introduce rainfall-runoff before abbreviating.
L 279 – define mOD before using abbreviation.
L 302 – grammar: either "which therefore leads us to conclude" or "which therefore indicates"
L 325 – missing space after Figure 5
L 494 – " property or human life"

**Additional References:**

- Brenner, S., Coxon, G., Howden, N. J. K., Freer, J., and Hartmann, A.: Process-based modelling to evaluate simulated groundwater levels and frequencies in a Chalk catchment in south-western England, Nat. Hazards Earth Syst. Sci., 18, 445–461
- Jackson CR, Bloomfield JP, Mackay JD. Evidence for changes in historic and future groundwater levels in the UK. Progress in Physical Geography: Earth and Environment. 2015;39(1):49-67.
- Flanagan, J., Nolan, P., McGrath, R. and Werner, C., (2019). Towards a definitive historical high-resolution climate dataset for Ireland – promoting climate research in Ireland. *Advances in Science and Research* 15: 263–276.
- Flanagan, J., Nolan, P. (2020) *Towards a Definitive Historical High-resolution Climate Dataset for Ireland – Promoting Climate Research in Ireland*. EPA Research 350. Available at https://www.epa.ie/pubs/reports/research/climate/researchreport350/ (accessed 11 November 2020).
- Irvine K., Coxon C., Gill L., Kimberley S., Waldren S. (2018). Chapter: Turloughs (Ireland). In: *The Wetland Book - I: Structure and Function, Management, and Methods* Finlayson, C.M., Everard, M., Irvine, K., McInnes, R., Middleton, B., van Dam, A., Davidson, N.C. (Eds.) Springer, Netherlands pp 1067-1077  [ISBN: 978-90-481-3493-9]

- Kimberley S., Naughton O., Johnston P.M., Gill L.W., Waldren S. (2012). The influence of flood duration on the surface soil properties and grazing management of karst wetlands (turloughs) in Ireland. *Hydrobiologia* 692, 29-40.
- McGrath, R. and Lynch, P. (eds), 2008. *Ireland in a Warmer World: Scientific Predictions of the Irish Climate in the Twenty-first Century. Community Climate Change Consortium for Ireland (C4I).* Available online: https://www.epa.ie/pubs/reports/research/climate/EPA_climate_change_regional_models_ERTDI36.pdf (accessed 11 November 2020).
- McGrath, R., Nishimura, E., Nolan, P., Semmler, T., Sweeney, C. and Wang, S., 2005. *Climate Change: Regional Climate Model Predictions for Ireland.* Environmental Protection Agency, Johnstown Castle, Ireland.
- Moss, R.H., Edmonds, J.A., Hibbard, K.A., Manning, M.R., Rose, S.K., van Vuuren, D.P., et al., 2010. The next generation of scenarios for climate change research and assessment. *Nature* 463(7282): 747–756.
- Mudarra, M., Hartmann, A., & Andreo, B. (2019). Combining experimental methods and modeling to quantify the complex recharge behavior of karst aquifers. Water Resources Research, 55, 1384– 1404.
- Nerantzaki, S. D. and Nikolaidis, N., P., (2020). The response of three Mediterranean karst springs to drought and the impact of climate change, Journal of Hydrology, 591 (125296)
- Nolan, P., Flanagan, J. (2020). *High-Resolution Climate Projections for Ireland – A Multi-model Ensemble Approach*. EPA Research Report, 339. Read: http://epa.ie/pubs/reports/research/climate/researchreport339/ (accessed 11 November 2020).
- Pardo-Igúzquiza, E., Collados-Lara, A.J. & Pulido-Velazquez, D., (2019). Potential future impact of climate change on recharge in the Sierra de las Nieves (southern Spain) high-relief karst aquifer using regional climate models and statistical corrections. Environ Earth Sci 78, 598
- Porst G., Irvine K. (2009). Distinctiveness of macroinvertebrate communities in turloughs (temporary ponds) and their response to environmental variables *Aquatic Conserv: Mar. Freshw. Ecosyst*.19: 456–465
- Porst G., Naughton O., Gill L., Johnston P., Irvine K. (2012). Adaptation, phenology and disturbance of macroinvertebrates in temporary water bodies. *Hydrobiologia* 696, 47-62.
- Nerantzaki, S. D., Hristopulos, D. T. and Nikolaidis, N. P, (2020). Estimation of the uncertainty of hydrologic predictions in a karstic Mediterranean watershed. Science of The Total Environment, 717.
- van Vuuren, D.P., Edmonds, J., Kainuma, M.L.T., Riahi, K., Thomson, A., Matsui, T., et al., 2011. The representative concentration pathways: an overview. *Climatic Change* 109(11): 5–31.
- Waldren S., Allott N., Coxon C., Cunha Periera H., Gill L., Gonzalez A., Irvine K., Johnston P., Kimberley S., Murphy M., Naughton O., O'Rourke A., Penck M., Porst G., Sharkey N. (2015). Turlough Hydrology, Ecology and Conservation. Unpublished Report, National Parks & Wildlife Services. Department of Arts, Heritage and the Gaeltacht, Dublin, Ireland.
- Werner C., Nolan P. and Naughton, O., 2019. High-resolution Gridded Datasets of Hydro-climate Indices for Ireland. Environmental Protection Agency, Johnstown Castle, Ireland.

---

## Author Response (AR2)

**Final review – author responses**

**L 78: Probably „models" not „modes" ?**

This error has been corrected.

**Figure 6: I believe the term „Peak Frequency" is not defined in the manuscript, which prevents interpretation of the figure. Could you add it?**

Text has been added to the image caption to clarify.

**Figure 7: Additionally to the comment by the reviewer, could you move the legend to the bottom right and increase the font size of the numbers? Currently it is difficult to read.**

This had been completed.

**Could you please carefully read the manuscript a last time. There are a couple of typos / gaps that are probably remaining from the revision procedure.**

Lead author has re-read and checked the document. These typos and gaps have been corrected.